# Rethinking Gradient Step Denoiser: Towards Truly Pseudo-Contractive Operator

**Shuchang Zhang**
College of Science
National University of Defense Technology
Changsha, 410073
zhangshuchang19@nudt.edu.cn

**Yaoyun Zeng**[*]
College of Science
National University of Defense Technology
Changsha, 410073
yaoyun_zeng@nudt.edu.cn

**Kangkang Deng**
College of Science
National University of Defense Technology
Changsha, 410073
freedeng1208@gmail.com

**Hongxia Wang** [†]
College of Science
National University of Defense Technology
Changsha, 410073
wanghongxia@nudt.edu.cn

## Abstract

Learning pseudo-contractive denoisers is a fundamental challenge in the theoretical analysis of Plug-and-Play (PnP) methods and the Regularization by Denoising (RED) framework. While spectral methods attempt to address this challenge using the power iteration method, they fail to guarantee the truly pseudo-contractive property and suffer from high computational complexity. In this work, we rethink gradient step (GS) denoisers and establish a theoretical connection between GS denoisers and pseudo-contractive operators. We show that GS denoisers, with the gradients of convex potential functions parameterized by input convex neural networks (ICNNs), can achieve truly pseudo-contractive properties. Furthermore, we integrate the learned truly pseudo-contractive denoiser into the RED-PRO (RED via fixed-point projection) model, definitely ensuring convergence in terms of both iterative sequences and objective functions. Extensive numerical experiments confirm that the learned GS denoiser satisfies the truly pseudo-contractive property and, when integrated into RED-PRO, provides a favorable trade-off between interpretability and empirical performance on inverse problems.

## 1 Introduction

The pseudo-contractive operators constitute a wide class of operators that arise in iterative methods for solving fixed point problems [17]. Here, we recall that an operator $T : \mathcal{H} \to \mathcal{H}$ is $d$-pseudo-contractive ($d$-PC) if there exists a constant $d \in (-\infty, 1]$ such that for any $\mathbf{x}, \mathbf{y} \in \mathcal{H}$, it holds that [12, 31]

$$\|T(\mathbf{x}) - T(\mathbf{y})\|^2 \le \|\mathbf{x} - \mathbf{y}\|^2 + d\|(\mathbf{x} - T(\mathbf{x})) - (\mathbf{y} - T(\mathbf{y}))\|^2, \tag{1}$$

where $\mathcal{H}$ is a Hilbert space. When $d < 1$, the operator $T$ is called $d$-strictly PC ($d$-SPC) operator [17]. The operator $T$ is nonexpansive and firmly nonexpansive (FNE) when $d = 0$ and $d = -1$ [6]. The pseudo-contractive assumption has played an important role in the convergence analysis of PnP methods [60, 53, 14, 51, 56, 59, 40, 47, 57, 48, 30, 4, 11, 61, 46, 52, 10] and RED

---

[*]Equal contribution
[†]Corresponding author

39th Conference on Neural Information Processing Systems (NeurIPS 2025).

framework [50, 49, 21, 42]. Firmly nonexpansive (FNE) denoisers [59, 48, 57, 11, 10], averaged non-expansive denoisers [56, 47, 30, 46], nonexpansive denoisers [53, 50, 14, 49, 40, 52], and contractive residuals or operators [51, 42, 35, 4] all belong to the class of pseudo-contractive operators [17]. The demicontractive assumption (see Definition 3.1) further generalizes the SPC concept, enabling the inclusion of a broader range of denoisers. Based on the fixed-point projection of demicontractive denoisers, Cohen et al. proposed the RED-PRO model [21] to theoretically bridge between PnP and RED priors. The indicator of the fixed-point set can serve as a regularizer and plays an increasingly important role in solving inverse problems. It is critically important to develop an efficient technique for testing the demicontractivity of a given mapping [21]. Of course, obtaining a pseudo-contractive denoiser also ensures the demicontractive property, as pseudo-contractive denoisers are a subclass of demicontractive operators shown in Figure 1. However, ensuring nonexpansivity, or more generally, a Lipschitz constraint on artificial neural networks is not easy in practice [48, 11]. Therefore, how to design an efficient training framework that enables neural networks to learn theoretically guaranteed denoising mappings under the weak assumption, such as (1), is a highly challenging problem. In this paper, we will try to answer the open question by establishing a theoretical connection between the GS denoiser and the pseudo-contractive denoiser. **The GS denoiser is all you need—simply parameterizing convex potential functions with ICNNs is sufficient without complex training techniques.**

Recently, spectral methods have been proposed to address the aforementioned problem [51, 48, 35, 61]. The SPC-DRUNet trained by spectral methods has achieved the state-of-the-art performance in image restoration problems [61]. Spectral methods typically use the power iteration (PI) method to compute the spectral norm of the Jacobian matrix, which is then either normalized by dividing the convolutional kernels by the spectral norm [51] or incorporated into the loss function as a penalty during training [48, 35, 61]. Ryu et al. [51] utilize a PI method, treating the convolution as a linear operator that performs a matrix-vector product. To balance speed and precision, PI only runs one iteration. They enforced the contractiveness of the residual $I - T_\theta$ of the denoiser $T_\theta$ by real spectral normalization (RealSN), which normalized the spectral norm of each layer. To obtain a FNE denoiser $T_\theta$, which is equivalent to the nonexpansiveness of $Q_\theta = 2T_\theta - I$ (see [16, Theorem 2.2.10] and [7, Proposition 4.4]), Pesquet et al. [48] added the penalty term of the spectral norm to the loss function, i.e.,

$$\mathcal{L}(\theta) = \mathbb{E}_{\mathbf{x}\sim p(\mathbf{x}),\mathbf{n}\sim\mathcal{N}(\mathbf{0},\sigma^2\mathbf{I})} \left\|T_\theta(\mathbf{x}+\mathbf{n}) - \mathbf{x}\right\|^2 + \lambda \max\{\left\|J_{Q_\theta}(\tilde{\mathbf{x}})\right\|_*, 1-\varepsilon\},$$

where $p(\mathbf{x})$ is the data distribution, $J_{Q_\theta}(\tilde{\mathbf{x}})$ is the Jacobian of $Q_\theta$ at point $\tilde{\mathbf{x}} = \varrho\mathbf{x} + (1-\varrho)T_\theta(\mathbf{x}+\mathbf{n})(\varrho \in [0,1])$, the noise $\mathbf{n}$ follows a Gaussian distribution with mean $\mathbf{0}$ and standard deviation $\sigma\mathbf{I}$, $\lambda > 0$ is a penalization parameter, and the parameter $\varepsilon \in (0,1)$ controls the penalty term. Then the denoiser $T_\theta$ is a resolvent of a maximally monotone operator (MMO). Later, the GS denoiser $T_\theta = I - \nabla\psi_\theta$ was proposed [20, 34], where $\psi_\theta : \mathbb{R}^n \to \mathbb{R} \cup \{\infty\}$ is parameterized by the differentiable neural network such as ICNN [2] and DRUNet [66]. Based on the GS denoiser, Hurault et al. proposed the proximal DRUNet (Prox-DRUNet) [35], which requires that $\nabla\psi_\theta$ is $L$-contractive with $L < 1$. They fine-tuned the previously trained GS denoiser by the spectral method with the following loss:

$$\mathcal{L}(\theta) = \mathbb{E}_{\mathbf{x}\sim p(\mathbf{x}),\mathbf{n}\sim\mathcal{N}(\mathbf{0},\sigma^2\mathbf{I})} \left\|T_\theta(\mathbf{x}+\mathbf{n}) - \mathbf{x}\right\|^2 + \lambda \max\{\left\|J_{\nabla\psi_\theta}(\mathbf{x}+\mathbf{n})\right\|_*, 1-\varepsilon\}.$$

During training, the spectral norm $\left\|J_{\nabla\psi_\theta}(\mathbf{x}+\mathbf{n})\right\|_*$ is estimated with 50 iterations. Wei et al. trained the $d$-SPC ($d < 1$) denoiser by the spectral method with the following loss [61]

$$\mathcal{L}(\theta) = \mathbb{E}_{\mathbf{x}\sim p(\mathbf{x}),\mathbf{n}\sim\mathcal{N}(\mathbf{0},\sigma^2\mathbf{I})} \left\|T_\theta(\mathbf{x}+\mathbf{n}) - \mathbf{x}\right\|^2 + \lambda \max\{\left\|dI + (1-d)J_{T_\theta}(\mathbf{x}+\mathbf{n})\right\|_*, 1-\varepsilon\}.$$

However, these spectral methods constrain the spectral norm using a finite number of training samples $\{\mathbf{x}_i, \mathbf{x}_i^*\}_{i=1}^{N_0}$ (noisy and clean image pairs), rather than all samples from the entire space. It is unable to accurately constrain the global spectral norm such as $\max_{\mathbf{x}\in\mathbb{R}^n} \left\|2J_{T_\theta}(\mathbf{x}) - I\right\|_*$ in [48], which may violate the SPC property. Moreover, PI methods have notable drawbacks: they can be computationally expensive [61], are not fully deterministic due to random initialization, and intermediate iterations do not guarantee a strict upper bound on the spectral norm [25].

Amos et al. [2] introduced ICNNs, which guarantee the convexity by imposing non-negative constraints on network weights and employing convex, non-decreasing activation functions. The gradients of ICNNs possess universal approximation capabilities [33], making them particularly valuable in data-driven optimal transport [44, 33]. The GS denoiser leverages ICNNs to learn gradients of convex implicit regularizers [20]. Fang et al. [28] further proposed learned proximal networks (LPN), which

are parameterized by the gradient of ICNN. In this work, we rethink the GS denoiser $T_\theta = I - \nabla \psi_\theta$, where $\psi_\theta$ is an ICNN, that can serve as a truly SPC denoiser. We are the first to establish the essential connection between SPC denoisers and GS denoisers. The main contributions of this work are summarized as follows:

1. **Theoretical contributions.** We rethink that the GS denoiser corresponds to a truly SPC denoiser, as demonstrated in Proposition 4.1. We also theoretically propose another novel construction method for truly pseudo-contractive denoisers, as presented in Proposition 4.3. This work is the first to theoretically confirm the feasibility of training truly SPC denoisers and to practically address the unresolved challenge of training demicontractive denoisers posed by the RED-PRO model [21].

2. **Algorithms and applications.** We integrate the learned truly SPC denoiser into the RED-PRO model and propose Algorithm 1 for solving imaging inverse problems. Theorem 4.6 further establishes the convergence of objective functions in RED-PRO, complementing the prior work [21] that focused solely on sequence convergence.

3. **Experimental validation.** Through extensive numerical experiments, we validate the SPC property of the GS denoiser compared to spectral methods [48, 61]. Our results highlight the ability of the method to balance interpretability and performance in addressing inverse problems.

## 2 Related works

Regularization models are important in image restoration problems, which can be formulated as follows:

$$\min_{\mathbf{x} \in \mathbb{R}^n} f(\mathbf{x}) + \lambda g(\mathbf{x}), \tag{2}$$

where $f$ is the data fidelity, $g$ is the regularizer, and $\lambda > 0$ is a regularization parameter. For example, the data fidelity $f(\mathbf{x}) = \frac{1}{2\sigma^2} \|\mathbf{A}\mathbf{x} - \mathbf{y}\|^2$ corresponds to $\mathbf{y} = \mathbf{A}\mathbf{x} + \mathbf{n}$ with given linear operator $\mathbf{A}$ and Gaussian noise $\mathbf{n}$.

### 2.1 PnP methods

PnP methods that combine splitting algorithms with denoiser priors have been widely applied in practical problems [60, 53, 1, 37, 62, 28, 63, 38] and have achieved state-of-the-art performance in inverse imaging tasks [66, 26, 35, 58]. Venkatakrishnan et al. [60] first proposed the PnP method using the alternating direction method of multipliers (ADMM). PnP methods solve the problem (2) by replacing the proximal operator $\mathrm{prox}_g(\mathbf{x})$ with denoisers, such as non-local means (NLM) [13] and block-matching 3D filtering (BM3D) [24], within ADMM or forward-backward splitting (FBS), also known as the proximal gradient method [9]. Zhang et al. [66] extended PnP methods using trained DNNs to achieve state-of-the-art performance in image restoration [66]. The convergence of PnP methods has been extensively studied. Sreehari et al. established theoretical conditions for PnP-ADMM, requiring that the Jacobian $\nabla D_\sigma$ be a doubly stochastic and symmetric matrix with all real eigenvalues in the range $(0, 1]$ [53]. Buzzard et al. provided a Consensus Equilibrium interpretation on denoiser priors [14]. Chen et al. [18] analyzed fixed-point convergence under bounded denoisers, while Ryu et al. [51] proved the fixed-point convergence of PnP-FBS and PnP-ADMM using the Banach contraction principle, assuming strongly convex data fidelity and nonexpansiveness of the residual of DnCNN [67]. Diffusion models (DMs) [32] can also act as efficient PnP priors, which have been widely used in physical sciences such as black hole imaging problems [63, 68], and image restoration [69].

### 2.2 RED framework

Romano et al. [50] introduced the well-known RED framework, which constructs an explicit objective function and flexibly incorporates various denoisers, such as NLM [13], BM3D [24], or trainable nonlinear reaction diffusion (TNRD) [19]. The RED framework has been widely used in computational imaging [54, 55, 41, 42]. Reehorst and Schniter et al. [49] highlighted that some existing denoisers do not satisfy the assumptions of RED, and provided the SMD (score-matching by denoising) interpretation. To explore the relationship between PnP priors and RED, the RED-PRO

framework was proposed from the perspective of fixed-point projection. Since the fixed-point set $\text{Fix}(T) = \{\mathbf{x} \in \mathcal{H} : T(\mathbf{x}) = \mathbf{x}\}$ is closed and convex [21, Theorem 3.8], RED-PRO reformulates RED as a convex optimization problem for image restoration via fixed-point projection, the regularizer $g$ in (2) is the indicator of the fixed-point set $\text{Fix}(T)$, i.e.,

$$g(\mathbf{x}) = \begin{cases} 0, & \text{if } \mathbf{x} \in \text{Fix}(T), \\ \infty, & \text{otherwise.} \end{cases}$$

Building on the idea of RED, the GS denoiser $T_\theta = I - \nabla \psi_\theta$ was proposed [20, 34], where $\psi_\theta$ is a differentiable function. Based on the GS denoiser, Hurault et al. [35] established the convergence theory of PnP methods. He et al. proposed simultaneous local and nonlocal RED (SLN-RED) for image restoration [29]. To avoid tuning the regularization parameter, Cascarano et al. proposed the constrained RED called CRED [15] based on the discrepancy principle [27].

# 3 Preliminaries

Cohen et al. first introduced the following demicontractive assumption on denoisers [21]. Here is the definition of demicontractive operators.

**Definition 3.1.** The mapping $T : \mathcal{H} \to \mathcal{H}$ is $d$-demicontractive with $d < 1$, if for any $\mathbf{x} \in \mathcal{H}$ and $\mathbf{z} \in \text{Fix}(T)$ it holds that

$$\|T(\mathbf{x}) - \mathbf{z}\|^2 \leq \|\mathbf{x} - \mathbf{z}\|^2 + d \|T(\mathbf{x}) - \mathbf{x}\|^2.$$

The $d$-SPC operator is a $d$-demicontractive operator.

Let $\mathcal{C}_1, \mathcal{C}_2, \mathcal{C}_3, \mathcal{C}_4$ denote the classes of all operators $T : \mathcal{H} \to \mathcal{H}$ satisfying the assumptions of demicontractive, SPC, NE, and FNE, respectively. For example, consider the inclusion defined as follows. Let $T \in \mathcal{C}_4$ be arbitrary. Then, for all $\mathbf{x}, \mathbf{y} \in \mathcal{H}$, it holds that

$$\|T(\mathbf{x}) - T(\mathbf{y})\|^2 \leq \|\mathbf{x} - \mathbf{y}\|^2 - \|(\mathbf{x} - T(\mathbf{x})) - (\mathbf{y} - T(\mathbf{y}))\|^2 \implies \|T(\mathbf{x}) - T(\mathbf{y})\| \leq \|\mathbf{x} - \mathbf{y}\|,$$

which means $T \in \mathcal{C}_3$. Therefore, $\mathcal{C}_4 \subset \mathcal{C}_3$. The relationship between different classes of operators is shown in Figure 1.



Figure 1: Relationship between different classes of operators.

The operator $T : \mathcal{H} \to \mathcal{H}$ is called conically $\lambda$-averaged for $\lambda > 0$ [5] if there exists a nonexpansive operator $U$ such that $T = (1 - \lambda)I + \lambda U$, where $I$ denotes the identity. In particular, when $\lambda \in (0, 1)$ the operator is $\lambda$-averaged, a class that plays an important role in fixed-point algorithms [22, 64, 65, 8, 23]. If $U$ is FNE, then $T = (1 - \lambda)I + \lambda U$ is $\lambda$-relaxed FNE ($\lambda$-RFNE).

Next, we give some equivalent relationships between $d$-SPC operators, conically averaged operators, and RFNE operators demonstrated in Proposition 3.2. We give the proof in Appendix A.

**Proposition 3.2** ([17]). *Let $\lambda = \frac{1}{1-d}$. Then the following statements are equivalent:*

*(i) Let $T$ be $d$-SPC ($d < 1$).*

*(ii) $T$ is a conically $\lambda$-averaged operator.*

*(iii) $T$ is a $2\lambda$-RFNE operator.*

The following Proposition 3.3 serves as a key connection for exploring how to learn a truly pseudo-contractive denoiser. Please see the proof in Appendix B.

**Proposition 3.3** ([16]). *Let $R = I - T$ and $\mu > 0$, then $T : \mathcal{H} \to \mathcal{H}$ is $\mu$-RFNE if and only if for all $\mathbf{x}, \mathbf{y} \in \mathcal{H}$,*

$$\langle \mathbf{x} - \mathbf{y}, R(\mathbf{x}) - R(\mathbf{y}) \rangle \geq \frac{1}{\mu} \| R(\mathbf{x}) - R(\mathbf{y}) \|^2. \tag{3}$$

The residual $R = I - T$ is also called $\frac{1}{\mu}$-cocoercive in (3). Figure 2 shows all equivalence relationships between $1 - \frac{1}{\lambda}$-SPC operator, conically $\lambda$-averaged operator, $2\lambda$-RFNE operator, and $\frac{1}{2\lambda}$-cocoercive residual, where $\lambda = \frac{1}{1-d}$.

$$\boxed{\begin{array}{c} R = I - T \text{ is a} \\ \frac{1}{2\lambda}\text{-cocoercive} \\ \text{operator} \end{array}}$$

$$\Updownarrow$$

$$\boxed{\begin{array}{c} T \text{ is a conically} \\ \lambda\text{-averaged operator} \end{array}} \quad \Leftrightarrow \quad \boxed{\begin{array}{c} T \text{ is a } 1 - \frac{1}{\lambda}\text{-SPC} \\ \text{operator} \end{array}} \quad \Leftrightarrow \quad \boxed{\begin{array}{c} T \text{ is a } 2\lambda\text{-RFNE} \\ \text{operator} \end{array}}$$

Figure 2: Equivalent relationships.

In the following Proposition 3.4, we introduce an important property of $\alpha$-averaged operators for the convergence rate of objective functions about the RED-PRO model. The proof is given in Appendix C.

**Proposition 3.4** ([65]). *Let $T$ be a $\alpha$-averaged operator with $\alpha \in (0,1)$, then $T$ is a $\frac{1-\alpha}{\alpha}$-strongly quasi-nonexpansive operator, i.e., for any $\mathbf{z} \in \text{Fix}(T)$, it holds that*

$$\| T(\mathbf{x}) - \mathbf{z} \|^2 \leq \| \mathbf{x} - \mathbf{z} \|^2 - \frac{1-\alpha}{\alpha} \| \mathbf{x} - T(\mathbf{x}) \|^2. \tag{4}$$

Finally, we give several equivalent characterizations of the $L$-smoothness property over the entire space $\mathcal{H}$. Here is the definition of $L$-smoothness.

**Definition 3.5** ([9]). *Let $L > 0$. The function $h : \mathcal{H} \to \mathbb{R} \cup \{\infty\}$ is $L$-smooth if it is differentiable over $\mathcal{H}$ and satisfies*

$$\| \nabla h(\mathbf{x}) - \nabla h(\mathbf{y}) \| \leq L \| \mathbf{x} - \mathbf{y} \|, \quad \forall \mathbf{x}, \mathbf{y} \in \mathcal{H}. \tag{5}$$

**Theorem 3.6** ([9]). *Let $h : \mathcal{H} \to \mathbb{R} \cup \{\infty\}$ be a convex function, differentiable over $\mathcal{H}$, and let $L > 0$. Then the following claims are equivalent:*

*(i) $h$ is $L$-smooth.*

*(ii) For all $\mathbf{x}, \mathbf{y} \in \mathcal{H}$,*

$$\langle \nabla h(\mathbf{x}) - \nabla h(\mathbf{y}), \mathbf{x} - \mathbf{y} \rangle \geq \frac{1}{L} \| \nabla h(\mathbf{x}) - \nabla h(\mathbf{y}) \|^2. \tag{6}$$

# 4 Learned truly SPC denoiser

## 4.1 The GS denoiser is all you need

In this section, we will prove that the GS denoiser $T_\theta = I - \nabla \psi_\theta$ in which $\psi_\theta$ is an ICNN, can precisely correspond to a truly SPC denoiser. In [20], although Cohen et al. proposed the GS denoiser early, they failed to explore its connection with the SPC denoiser. The pursuit of learning SPC denoisers, once perceived as distant, is now within reach. Two inequalities (3) and (6) form a crucial bridge, enabling us to establish the essential connection between the GS denoiser and the SPC denoiser. We give the following result to theoretically guarantee the SPC property of the GS denoiser.

**Proposition 4.1.** *Consider a scalar-valued $(K+1)$-layered neural network $\psi_\theta : \mathbb{R}^n \to \mathbb{R}$ defined by $\psi_\theta(\mathbf{x}) = \mathbf{w}^{\mathrm{T}} \mathbf{z}_K + b$ and the recursion*

$$\mathbf{z}_1 = \phi(\mathbf{H}_1 \mathbf{x} + \mathbf{b}_1), \quad \mathbf{z}_k = \phi(\mathbf{W}_k \mathbf{z}_{k-1} + \mathbf{H}_k \mathbf{x} + \mathbf{b}_k), \quad k = 2, 3, \ldots, K,$$

*where $\Theta = \{\mathbf{w}, b, \{\mathbf{W}_k\}_{k=2}^{K}, \{\mathbf{H}_k\}_{k=1}^{K}, \{\mathbf{b}_k\}_{k=1}^{K}\}$ are learnable parameters, $\phi : \mathbb{R} \to \mathbb{R}$ is a convex, non-decreasing and continuously differentiable scalar function, which operates pointwise. Assume that all entries of $\mathbf{W}_k$ and $\mathbf{w}$ are non-negative, and let $\psi_\theta$ be $L_\theta$-smooth, then the GS denoiser $T_\theta = I - \nabla \psi_\theta$ is $\frac{L_\theta - 2}{L_\theta}$-SPC operator.*

*Proof.* Since $\mathbf{W}_k$ and $\mathbf{w}$ are non-negative, it follows that $\psi_\theta$ is convex from [2, Proposition 1]. Since $\psi_\theta$ is $L_\theta$-smooth, by (6), we have

$$\langle \nabla \psi_\theta(\mathbf{x}) - \nabla \psi_\theta(\mathbf{y}), \mathbf{x} - \mathbf{y} \rangle \geq \frac{1}{L_\theta} \|\nabla \psi_\theta(\mathbf{x}) - \nabla \psi_\theta(\mathbf{y})\|^2.$$

Let $T_\theta = I - \nabla \psi_\theta$, by Proposition 3.2 and recall (3) in Proposition 3.3, we directly derive $\frac{2}{1-d} = L_\theta$, i.e., $d = \frac{L_\theta - 2}{L_\theta}$. Therefore, the denoiser $T_\theta$ is a $\frac{L_\theta - 2}{L_\theta}$-SPC operator, which completes the proof. $\square$

Given the ICNN $\psi_\theta$, we train the GS denoiser $T_\theta = I - \nabla \psi_\theta$ with the following loss function

$$\mathcal{L}(\theta) = \mathbb{E}_{\mathbf{x} \sim p(\mathbf{x}), \mathbf{n} \sim \mathcal{N}(\mathbf{0}, \sigma^2 \mathbf{I})} \|T_\theta(\mathbf{x} + \mathbf{n}) - \mathbf{x}\|^2.$$

As previously proposed in [20], the training process is straightforward and does not require any additional spectral norm penalty.

Our Proposition 4.1 is the first to realize a truly $\frac{L_\theta - 2}{L_\theta}$-SPC operator via the ICNN GS denoiser, whose assumption is weaker than FNE and nonexpansive, and thus easier to satisfy in practice. Once the GS denoiser meets the $\frac{L_\theta - 2}{L_\theta}$-SPC condition, the RED-PRO framework automatically guarantees sequence convergence and objective convergence rate, as shown in Theorems 4.5 and 4.6, without requiring stronger FNE or nonexpansive assumptions. The result of Proposition 4.1 can further benefit existing PnP/RED theoretical works by enabling the ICNN GS denoiser to satisfy stronger FNE or averaged nonexpansive assumptions in two ways:

- Controlling $L_\theta \leq 1$, e.g., by normalizing convolution kernels via spectral methods or penalizing the network's Lipschitz constant in the loss function, so that the FNE and nonexpansive assumptions required in [57, Assumption 2], [53, Theorem III.1], and [49, Lemma 5] are met;

- Estimating $L_\theta$ via the power method and tuning the weight $w < \frac{2}{L_\theta}$ so that $T_w = w T_\theta + (1-w)I$ is a $\frac{w L_\theta}{2}$-averaged operator, thus satisfying the averaged operator assumptions in [56, Assumption 2(b)] and [47, Theorem 3.5, Theorem 3.6]

Therefore, Proposition 4.1 provides valuable practical guidance for existing PnP/RED theoretical works that require FNE, and averaged nonexpansive assumption.

*Remark* 4.2. Although we use the same GS denoiser as in [20], the difference is that we are the first to theoretically address the relationship between $d$-SPC denoisers and GS denoisers, and also practically demonstrate the feasibility of training SPC denoisers that fully satisfy the demicontractive condition required by RED-PRO [21]. We believe that this theoretical finding is valuable.

In contrast to spectral methods [51, 48, 35, 61], our approach, like [20], directly embeds the underlying mathematical structure, i.e., (3) and (6), into the denoiser, thereby naturally satisfying the SPC property. This eliminates the need for adding extra penalty terms in the loss function and overcomes the limitation that spectral methods cannot constrain the global spectral norm.

However, as pointed out in [20], one limitation is that the non-negative weights may constrain the expressivity of ICNNs. We are now theoretically give another alternative construction method for any truly pseudo-contractive neural networks.

**Proposition 4.3.** *Let $R_\theta : \mathbb{R}^n \to \mathbb{R}^n$ be a $L_\theta$-Lipschitz continuous convolutional neural networks. Denote $\tilde{R}_\theta = R_\theta + L_\theta I$, then the denoiser $T_\theta = I - \tilde{R}_\theta$ is a pseudo-contractive operator, i.e.,*

$$\|T_\theta(\mathbf{x}) - T_\theta(\mathbf{y})\|^2 \leq \|\mathbf{x} - \mathbf{y}\|^2 + \|(\mathbf{x} - T_\theta(\mathbf{x})) - (\mathbf{y} - T_\theta(\mathbf{y}))\|^2.$$

*Proof.* Since $R_\theta$ is $L_\theta$-Lipschitz continuous, for any $\mathbf{x}, \mathbf{y} \in \mathcal{H}$ we have

$$\|R_\theta(\mathbf{x}) - R_\theta(\mathbf{y})\| \le L_\theta \|\mathbf{x} - \mathbf{y}\|, \tag{7}$$

by (7) and Cauthy-Schwarz inequality, we have

$$\|\langle R_\theta(\mathbf{x}) - R_\theta(\mathbf{y}), \mathbf{x} - \mathbf{y}\rangle\| \le L_\theta \|\mathbf{x} - \mathbf{y}\|^2.$$

We can derive that

$$\langle R_\theta(\mathbf{x}) - R_\theta(\mathbf{y}), \mathbf{x} - \mathbf{y}\rangle + L_\theta \|\mathbf{x} - \mathbf{y}\|^2 \ge 0.$$

Thus, we have

$$\langle \tilde{R}_\theta(\mathbf{x}) - \tilde{R}_\theta(\mathbf{y}), \mathbf{x} - \mathbf{y}\rangle = \langle R_\theta(\mathbf{x}) - R_\theta(\mathbf{y}), \mathbf{x} - \mathbf{y}\rangle + L_\theta \|\mathbf{x} - \mathbf{y}\|^2 \ge 0.$$

By [7, Example 20.8], it follows that $T_\theta = I - \tilde{R}_\theta$ is a pseudo-contractive operator. $\square$

The core problem in Proposition 4.3 is how to training Lipschitz-constrained neural networks. Ryu et al. normalized each convolutional kernel $\mathcal{K}$ with estimated spectral norm $\|\mathcal{K}\|_*$, i.e., $\frac{\mathcal{K}}{\|\mathcal{K}\|_*}$, which can obtain 1-Lipschitz CNN. Anil et al. [3] proposed that by combining GroupSort activation functions with orthonormal weight matrices, one can construct networks that are provably 1-Lipschitz and capable of approximating any 1-Lipschitz function arbitrarily well. These methods can be used to train the 1-Lipschitz-constrained neural networks $R_\theta$ in Proposition 4.3. In this case, the Lipschitz constant $L_\theta$ is equal to 1, then $\tilde{R}_\theta = R_\theta + I$, and $\tilde{R}_\theta = R_\theta + I$ can be viewed as a residual connection, which is used to fit the noise distribution $\mathbf{n}$. That is, $\tilde{R}_\theta$ is obtained by minimizing the following loss function:

$$\min_\theta \left\{ \mathcal{L}(\theta) = \mathbb{E}_{\mathbf{x} \sim p(\mathbf{x}), \mathbf{n} \sim \mathcal{N}(\mathbf{0}, \sigma^2 \mathbf{I})} \left\| \tilde{R}_\theta(\mathbf{x} + \mathbf{n}) - \mathbf{n} \right\|^2 \right\},$$

and the pseudo-contractive denoiser $T_\theta = I - \tilde{R}_\theta$ is constructed. Moreover, Delattre et al. [25] controlled a $L$-Lipschitz convolutional kernel $\mathcal{K}_j (1 \le j \le l)$, the training loss becomes: $\mathcal{L}(\theta) + \mu_{\text{reg}} \sum_{j=1}^l \mathcal{L}_{\text{reg}}(\mathcal{K}_j)$ with

$$\mathcal{L}_{\text{reg}}(\mathcal{K}_j) = \sigma_{\text{GI}}(\mathcal{K}_j) \mathbf{1}_{\sigma_{\text{GI}}(\mathcal{K}_j) > L},$$

where $\sigma_{\text{GI}}$ denotes the spectral norm computed by Gram iteration (GI), which is more efficient and accurate than the power method, and $x \to \mathbf{1}_{x > L}$ indicates 1 if $x > L$, and 0 otherwise.

Although the above methods yields pseudo-contractive denoisers by training Lipschitz-constrained neural networks, such networks may practically suffer from limited expressive capacity [3, 36].

*Remark* 4.4. According to Proposition 4.3, as long as we accurately compute the Lipschitz constant of the neural network, we can construct the pseudo-contractive neural network. The recent work [25] has shown that it is feasible to accurately obtain the Lipschitz constant of neural networks. We believe that more expressive truly pseudo-contractive neural networks with inherent interpretability shown in Proposition 4.3 will be developed in the future.

## 4.2 RED-PRO with the learned SPC denoiser

Based on the fixed-point projection of the demicontractive denoiser, Cohen et al. proposed the following RED-PRO model

$$\min_{\mathbf{x} \in \text{Fix}(T_\theta)} f(\mathbf{x}), \tag{8}$$

where the denoiser $T_\theta$ is assumed to be $d$-demicontractive ($d < 1$) and $f(\mathbf{x}) = \frac{1}{2\sigma^2} \|\mathbf{A}\mathbf{x} - \mathbf{y}\|^2$. The hybrid steepest descent algorithms [64, 65] is used to solve (8), i.e.,

$$\mathbf{z}^k = T_w(\mathbf{z}^{k-1} - \mu_k \nabla f(\mathbf{z}^{k-1})), \tag{9}$$

where $T_w = w T_\theta + (1 - w)I$, and $\{\mu_k\}_{k \in \mathbb{N}}$ is diminishing, i.e., $\sum_{k \in \mathbb{N}} \mu_k = +\infty, \lim_{k \to \infty} \mu_k = 0$, and $w \in (0, \frac{1-d}{2})$ for $d$-demicontractive denoiser $T$. Proposition 3.2 shows that $T_w$ is $w\theta$-averaged and thus always nonexpansive for $0 < w < 1 - d$. Algorithm 1 shows the detailed steps of RED-PRO with the learned truly SPC denoiser $T_\theta$, which is written into the following compact form,

$$\mathbf{x}^k = T_w(\mathbf{x}^{k-1}) - \mu_k \nabla f(T_w(\mathbf{x}^{k-1})), \tag{10}$$

where $T_w = wT_\theta + (1-w)I$. In fact, the above iteration (10) is equivalent to the iteration (9).

---

**Algorithm 1** RED-PRO with the learned truly SPC denoiser

---

**Require:** initialization $\mathbf{x}^0 \in \mathbb{R}^n, \mu_k = \frac{c}{(1+k)^\alpha}, w \in (0, \frac{2}{L_\theta})$, and the GS denoiser $T_\theta = I - \nabla\psi_\theta$.
  1: **for** $k = 1, 2, \cdots, K$ **do**
  2:      $\mathbf{y}^k = (1-w)\mathbf{x}^{k-1} + wT_\theta(\mathbf{x}^{k-1})$
  3:      $\mathbf{x}^k = \mathbf{y}^k - \mu_k\nabla f(\mathbf{y}^k)$
  4: **end for**
**Ensure:** $\mathbf{x}^K$.

---

The known Theorem 4.5 provides the convergence guarantee of Algorithm 1 to an optimal solution of (2) with the learned truly $d$-SPC denoiser $T_\theta$. Compared to RED-PRO [21, Theorem 4.3], we can extend the interval of $w$ from $(0, \frac{1-d}{2})$ to $(0, 1-d)$, and explore that $w$ depends on the $L_\theta$-smooth property of the ICNN. We further complement the convergence rate of the objective function for RED-PRO in Theorem 4.6. The proof is given in the Appendix D.

**Theorem 4.5** ([65, 21]). *Let $T_\theta = I - \nabla\psi_\theta$ be a continuous $d$-SPC denoiser, and $f(\mathbf{x})$ be a proper convex l.s.c. differentiable function with L-Lipschitz gradient. Then the sequence $\{\mathbf{x}^k\}_{k\in\mathbb{N}}$ generated by Algorithm 1 converges to a solution of* (8).

**Theorem 4.6.** *Let $\{\mathbf{x}^k\}_{k\in\mathbb{N}}$ and $\{\mathbf{y}^k\}_{k\in\mathbb{N}}$ be sequences generated by Algorithm 1. Assume that $\mathcal{S} = \arg\min_{\mathbf{x}\in\text{Fix}(T_\theta)} f(\mathbf{x})$ is the solution set of the RED-PRO model and the sequence $\{\mathbf{x}^k\}_{k\in\mathbb{N}}$ is bounded, then*

  *(i) For any $\mathbf{x}' \in \mathcal{S}$ and $k \geq 1$, there exist $D_1, D_2 > 0$ such that*

$$u_k \leq \frac{D_1^2}{ck^{1-\alpha}} + \frac{cD_2^2}{k^\alpha},$$

  *where $u_k = \min\{\langle\nabla f(\mathbf{y}^j), \mathbf{y}^j - \mathbf{x}'\rangle : k \leq j \leq 2k\}$.*

  *(ii) For any $\mathbf{x}' \in \mathcal{S}$ and $k \geq 1$, we have*

$$f(\mathbf{y}_{best}^k) - f(\mathbf{x}') \leq \frac{D_1^2}{ck^{1-\alpha}} + \frac{cD_2^2}{k^\alpha},$$

  *where $k_{best} = \arg\min_{k\leq j\leq 2k} f(\mathbf{y}^j)$.*

*Remark* 4.7. The boundedness of $\{\mathbf{x}^k\}_{k\in\mathbb{N}}$ in Theorem 4.6 is straightforward to verify. Specifically, we can replace $T_w$ with $P_{[0,1]^n} \circ T_w$ in Algorithm 1, where $P_{[0,1]^n}$ denotes the metric projection onto the unit hypercube $[0, 1]^n$. Thus the sequence $\{\mathbf{y}^k\}_{k\in\mathbb{N}}$ is bounded such that $\mathbf{y}^k \in [0, 1]^n$. Since $\mathbf{x}^k = \mathbf{y}^k - \mu_k\nabla f(\mathbf{y}^k)$, for any $\mathbf{z} \in \text{Fix}(T)$, we have $\|\mathbf{x}^k - \mathbf{z}\| \leq \|\mathbf{y}^k - \mathbf{z}\| + c\|\nabla f(\mathbf{y}^k)\|$. Therefore, the sequence $\{\mathbf{x}^k\}_{k\in\mathbb{N}}$ is also bounded.

*Remark* 4.8. Compared to the convergence results of RED-PRO in [21, Theorem 4.3 and Theorem 4.4], we provide a new complementary analysis by establishing the outer convergence rate of the objective function in Theorem 4.6 (ii).

## 5 Experiments

In this section, we present some experiments to evaluate the performance of the learned truly SPC denoiser. We benchmark it against state-of-the-art spectral methods, including MMO [48] and SPCNet [61]. Specifically, we validate the SPC property of the learned denoiser. RED-PRO with the learned truly SPC denoiser has the theoretical guarantee, which can be applied to complicated imaging inverse problems. Our primary focus is on achieving theoretical interpretability rather than pursuing state-of-the-art performance.

### 5.1 Implementation details

We compare spectral methods with the DnCNN architecture. SPCNet [61] adopts $d = 0.5$ for MNIST and CelebA datasets [39, 43], and $d = 0.8$ for BSD400 [45], while the MMO method [48] uses

$d = -1$ in (1). Spectral methods are configured with $\lambda = 10^{-3}$, $\varepsilon = 0.1$, and 20 PI iterations for CelebA and BSD400, or 30 iterations for MNIST. The ICNN models start with an initial learning rate of $10^{-3}$, decaying to $10^{-4}$ after half the epochs. The DnCNN models begin with $10^{-4}$, decaying to $5 \times 10^{-5}$ mid-training.

For the MNIST dataset, ICNN is implemented with four convolutional layers, each containing 64 hidden neurons and softplus activation function $\phi(x) = \frac{1}{\beta} \log(1 + e^{\beta x})$ with $\beta = 10$. For BSD400 and CelebA, ICNN uses 256 hidden neurons with $\beta = 100$. All models are trained for Gaussian denoising with a noise level of $\sigma = 5/255$ and a batch size of 128. Training spans 50 epochs for MNIST and BSD400, and 30 epochs for CelebA. All experiments are conducted on one NVIDIA A800 GPU using the PyTorch framework.

## 5.2 Validation of SPC property

We test the SPC property of the learned GS denoiser $T_\theta = I - \nabla \psi_\theta$ with ICNN $\psi_\theta$, MMO [48], and SPCNet [61] on two MNIST and test12 datasets. We calculate the maximum $\hat{d}$ defined by

$$\hat{d} = \frac{\|T(\mathbf{x}) - T(\mathbf{y})\|^2 - \|\mathbf{x} - \mathbf{y}\|^2}{\|(\mathbf{x} - T(\mathbf{x})) - (\mathbf{y} - T(\mathbf{y}))\|^2},$$

where $\mathbf{y} = \mathbf{x} + \mathbf{n}, \mathbf{n} \sim \mathcal{N}(0, \sigma^2 \mathbf{I})$. Noise levels are uniformly sampled at 11 points in the interval $[10^{-5}, 10^{-2}]$. As shown in Figure 3, the SPCNet [61] obtains a maximum $\hat{d}$ that exceeds 0.8 and 0.5 at a noise level of $10^{-5}$, whereas the MMO method [48] yields a $\hat{d}$ value that exceeds $-1$. Therefore, denoisers trained by spectral methods violate the SPC property.

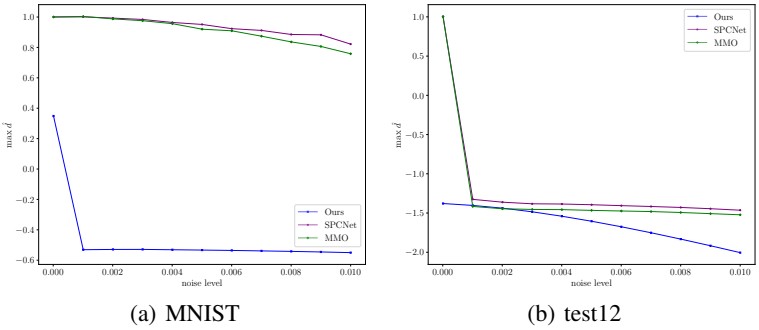

(a) MNIST             (b) test12

Figure 3: Validation of the truly SPC property on two different datasets.

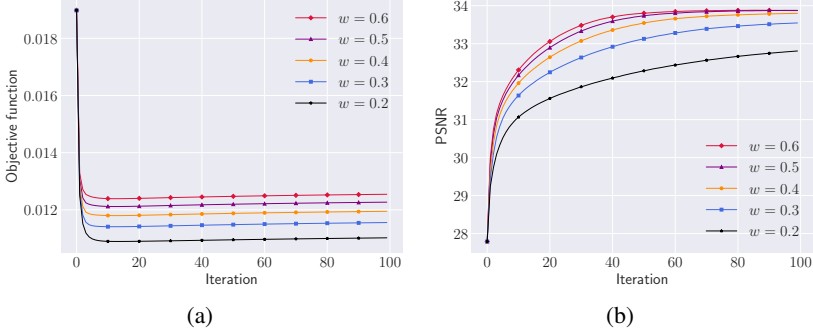

(a)             (b)

Figure 4: Convergence of Algorithm 1 on one image in the CelebA dataset. (a) Objective function. (b) PSNR.

## 5.3 Competitive results with theoretical guarantees

In Figure 4, we show the trend of the fidelity term $\frac{1}{2\sigma^2} \|\mathbf{y} - \mathbf{Ax}\|^2$ and PSNR (dB) throughout the iterations with different $w$. We provide additional results in Appendix E.

In the following, we demonstrate the effectiveness of RED-PRO with the SPC denoiser in inverse problems, highlighting the ability to achieve competitive results while strictly satisfying theoretical constraints. Compared to non-SPC methods such as DPIR [66], which achieve high PSNR in only 8 iterations but do not converge with more iterations (see Appendix G.5 in [35]). As shown in Table 1. RED-PRO can offer both competitive performance and guaranteed convergence. We also provide visual PSNR curves shown in Figure 5 to clearly demonstrate that non-SPC methods may not converge.

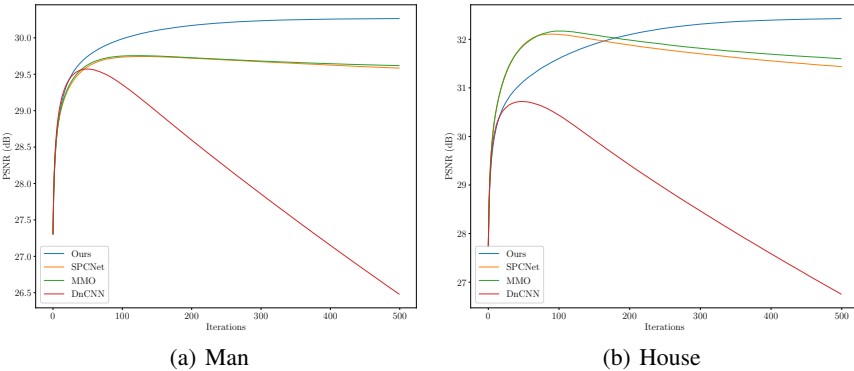

(a) Man    (b) House

Figure 5: PSNR curves of RED-PRO with the learned truly SPC denoiser and non-SPC methods on two images in the Gaussian deblurring task.

Table 1: Comparison of RED-PRO with various non-SPC denoisers on the Gaussian deblurring task. All non-SPC and the ICNN GS denoisers use Algorithm 1 with the same hyperparameters.

| Methods | Parrot | House | Boat | Couple | Man |
|---|---|---|---|---|---|
| SPC-DnCNN [61] | 25.73 | 31.43 | 28.53 | 28.17 | 29.58 |
| MMO [48] | 25.87 | 31.58 | 28.57 | 28.27 | 29.60 |
| DnCNN [67] | 25.30 | 26.74 | 26.23 | 25.98 | 26.47 |
| DRUNet [66] | 27.00 | 30.44 | 29.15 | 28.64 | 29.85 |
| GS denoiser [35] | **27.38** | **32.58** | 29.41 | 29.08 | 29.91 |
| Ours | 27.17 | 32.40 | **29.54** | **29.17** | **30.25** |

## 6  Conclusion

In this paper, we proposed a novel perspective to construct a truly SPC denoiser by directly embedding the underlying mathematical structure into the neural network architecture. Our theoretical analysis shown in Proposition 4.1 rethink that the known GS denoiser [20], built upon an ICNN, definitely satisfies the $d$-SPC property, which plays a crucial role in convergence in PnP methods and RED framework. Unlike spectral methods that require additional penalty terms and suffer from high computational cost due to costly PI iterations, our method naturally guarantees interpretability while offering the advantage in terms of time complexity. Furthermore, our theoretical insight shown in Proposition 4.3 can pave the way for future research in developing interpretative neural networks for imaging inverse problems.

## Acknowledgments

We sincerely thank the Associate Professor Hui Zhang for valuable discussions. We sincerely thank four anonymous reviewers for their valuable and constructive feedback, which has greatly improved our work. This work was supported by the following grants: the National Key Research and Development Program of China (No. 2020YFA0713504), the National Natural Science Foundation of China (Grant No. 12471401), and the National Natural Science Foundation of China — Young Scientists Fund (Grant No. 12401419).

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

# A The proof of Proposition 3.2

Let $\lambda = \frac{1}{1-d}$. We first show the equivalence between conically $\lambda$-averaged and $d$-SPC operators. Let $S = (1-d)T + dI$, then

$$
\begin{aligned}
\|S(\mathbf{x}) - S(\mathbf{y})\|^2 &= \|(1-d)(T(\mathbf{x}) - T(\mathbf{y})) + d(\mathbf{x} - \mathbf{y})\|^2 \\
&= (1-d)\|T(\mathbf{x}) - T(\mathbf{y})\|^2 + d\|\mathbf{x} - \mathbf{y}\|^2 \\
&\quad - d(1-d)\|(\mathbf{x} - T(\mathbf{x})) - (\mathbf{y} - T(\mathbf{y}))\|^2
\end{aligned}
$$

$$
\begin{aligned}
S \text{ is nonexpansive} &\Leftrightarrow \|S(\mathbf{x}) - S(\mathbf{y})\|^2 \leq \|\mathbf{x} - \mathbf{y}\|^2 \\
&\Leftrightarrow (1-d)\|T(\mathbf{x}) - T(\mathbf{y})\|^2 + d\|\mathbf{x} - \mathbf{y}\|^2 \\
&\quad - d(1-d)\|(\mathbf{x} - T(\mathbf{x})) - (\mathbf{y} - T(\mathbf{y}))\|^2 \leq \|\mathbf{x} - \mathbf{y}\|^2 \\
&\Leftrightarrow \|T(\mathbf{x}) - T(\mathbf{y})\|^2 \leq \|\mathbf{x} - \mathbf{y}\|^2 + d\|(\mathbf{x} - T(\mathbf{x})) - (\mathbf{y} - T(\mathbf{y}))\|^2 \\
&\Leftrightarrow T \text{ is } d\text{-SPC.}
\end{aligned}
$$

By the above equivalence, we obtain $T = (1-\lambda)I + \lambda S$ is conically $\lambda$-averaged, where $S$ is nonexpansive.

(i) $\Leftrightarrow$ (ii): Take $d = -1$, then the FNE operator $V$ is equivalent to $\frac{1}{2}$-averaged operator. There exists a NE operator $S$ such that $V = \frac{S+I}{2}$, then $S = 2V - I$. Substituting $S = 2V - I$ into $T = (1-\lambda)I + \lambda S$ yields

$$
T = (1-2\lambda)I + 2\lambda V,
$$

hence $T$ is $(2\lambda)$-relaxed FNE (i.e. $2\lambda$-RFNE). The conically $\lambda$-averaged operator is equivalent to the $2\lambda$-FNE operator.

# B The proof of Proposition 3.3

Let $T = \mu S + (1-\mu)I$ for a FNE operator $S$. Since $S$ is FNE, then

$$
\|S(\mathbf{x}) - S(\mathbf{y})\|^2 \leq \|\mathbf{x} - \mathbf{y}\|^2 - \|(\mathbf{x} - S(\mathbf{x})) - (\mathbf{y} - S(\mathbf{y}))\|^2.
$$

Since

$$
\begin{aligned}
\|S(\mathbf{x}) - S(\mathbf{y})\|^2 &= \|(S(\mathbf{x}) - \mathbf{x}) + (\mathbf{x} - \mathbf{y}) + (\mathbf{y} - S(\mathbf{y}))\|^2 \\
&= \|S(\mathbf{x}) - \mathbf{x}\|^2 + \|\mathbf{x} - \mathbf{y}\|^2 + \|\mathbf{y} - S(\mathbf{y})\|^2 \\
&\quad + 2\langle S(\mathbf{x}) - \mathbf{x}, \mathbf{x} - \mathbf{y}\rangle + 2\langle S(\mathbf{x}) - \mathbf{x}, \mathbf{y} - S(\mathbf{y})\rangle + 2\langle \mathbf{x} - \mathbf{y}, \mathbf{y} - S(\mathbf{y})\rangle.
\end{aligned}
$$

and

$$
\|(\mathbf{x} - S(\mathbf{x})) - (\mathbf{y} - S(\mathbf{y}))\|^2 = \|\mathbf{x} - S(\mathbf{x})\|^2 - 2\langle \mathbf{x} - S(\mathbf{x}), \mathbf{y} - S(\mathbf{y})\rangle + \|\mathbf{y} - S(\mathbf{y})\|^2.
$$

From

$$
\|S(\mathbf{x}) - S(\mathbf{y})\|^2 \leq \|\mathbf{x} - \mathbf{y}\|^2 - \|(\mathbf{x} - S(\mathbf{x})) - (\mathbf{y} - S(\mathbf{y}))\|^2,
$$

we obtain

$$
\begin{aligned}
\|\mathbf{x} - S(\mathbf{x})\|^2 + \|\mathbf{y} - S(\mathbf{y})\|^2 &\leq \langle \mathbf{x} - S(\mathbf{x}), \mathbf{y} - S(\mathbf{y})\rangle - \langle S(\mathbf{x}) - \mathbf{x}, \mathbf{x} - \mathbf{y}\rangle \\
&\quad - \langle S(\mathbf{x}) - \mathbf{x}, \mathbf{y} - S(\mathbf{y})\rangle - \langle \mathbf{x} - \mathbf{y}, \mathbf{y} - S(\mathbf{y})\rangle \\
&= \langle S(\mathbf{y}) - \mathbf{y}, S(\mathbf{x}) - \mathbf{x}\rangle - \langle \mathbf{x} - \mathbf{y}, S(\mathbf{x}) - \mathbf{x}\rangle \\
&\quad + \langle S(\mathbf{x}) - \mathbf{x}, S(\mathbf{y}) - \mathbf{y}\rangle + \langle \mathbf{x} - \mathbf{y}, S(\mathbf{y}) - \mathbf{y}\rangle \\
&= \langle S(\mathbf{y}) - \mathbf{x}, S(\mathbf{x}) - \mathbf{x}\rangle + \langle S(\mathbf{x}) - \mathbf{y}, S(\mathbf{y}) - \mathbf{y}\rangle,
\end{aligned}
$$

i.e.,

$$
\langle S(\mathbf{y}) - \mathbf{x}, S(\mathbf{x}) - \mathbf{x}\rangle + \langle S(\mathbf{x}) - \mathbf{y}, S(\mathbf{y}) - \mathbf{y}\rangle \geq \|S(\mathbf{x}) - \mathbf{x}\|^2 + \|S(\mathbf{y}) - \mathbf{y}\|^2. \quad (11)
$$

Since $S(\mathbf{x}) - \mathbf{x} = \frac{1}{\mu}(T(\mathbf{x}) - \mathbf{x})$ and $S(\mathbf{y}) - \mathbf{y} = \frac{1}{\mu}(T(\mathbf{y}) - \mathbf{y})$, then (11) is equivalent to

$$
\langle S(\mathbf{y}) - \mathbf{x}, T(\mathbf{x}) - \mathbf{x}\rangle + \langle S(\mathbf{x}) - \mathbf{y}, T(\mathbf{y}) - \mathbf{y}\rangle \geq \frac{1}{\mu}\left(\|T(\mathbf{x}) - \mathbf{x}\|^2 + \|T(\mathbf{y}) - \mathbf{y}\|^2\right). \quad (12)
$$

Let $R = I - T$, then
$$\langle S(\mathbf{y}) - \mathbf{x}, T(\mathbf{x}) - \mathbf{x} \rangle = \langle (S(\mathbf{y}) - \mathbf{y}) + (\mathbf{y} - \mathbf{x}), T(\mathbf{x}) - \mathbf{x} \rangle$$
$$= \langle \mathbf{x} - \mathbf{y}, R(\mathbf{x}) \rangle + \frac{1}{\mu} \langle R(\mathbf{x}), R(\mathbf{y}) \rangle. \tag{13}$$

Similarly, we have
$$\langle S(\mathbf{x}) - \mathbf{y}, T(\mathbf{y}) - \mathbf{y} \rangle = \langle \mathbf{y} - \mathbf{x}, R(\mathbf{y}) \rangle + \frac{1}{\mu} \langle R(\mathbf{x}), R(\mathbf{y}) \rangle. \tag{14}$$

By (12) and above two equations (13) and (14), we have
$$\langle \mathbf{x} - \mathbf{y}, R(\mathbf{x}) - R(\mathbf{y}) \rangle \geq \frac{1}{\mu} \| R(\mathbf{x}) - R(\mathbf{y}) \|^2.$$

## C   The proof of Proposition 3.4

Since $T$ is $\alpha$-averaged, by Proposition 3.2 (ii), then we have $\frac{1}{1-d} = \alpha \implies d = \frac{\alpha - 1}{\alpha}$, i.e., the operator $T$ is a $-\frac{1-\alpha}{\alpha}$-SPC operator,
$$\| T(\mathbf{x}) - T(\mathbf{y}) \|^2 \leq \| \mathbf{x} - \mathbf{y} \|^2 - \frac{1 - \alpha}{\alpha} \| (\mathbf{x} - T(\mathbf{x})) - (\mathbf{y} - T(\mathbf{y})) \|^2. \tag{15}$$

Let $\mathbf{y} = \mathbf{z} \in \text{Fix}(T)$ in (15), we finally obtain
$$\| T(\mathbf{x}) - \mathbf{z} \|^2 \leq \| \mathbf{x} - \mathbf{z} \|^2 - \frac{1 - \alpha}{\alpha} \| T(\mathbf{x}) - \mathbf{x} \|^2.$$

## D   The proof of Theorem 4.6

*Proof.* (i) For any $\mathbf{z} \in \text{Fix}(T)$, since $\mathbf{y}^k = T_w(\mathbf{x}^{k-1})$ and $T_w$ is $\frac{w}{1-d}$-averaged, by (4) we have
$$\left\| \mathbf{y}^k - \mathbf{z} \right\|^2 \leq \left\| \mathbf{x}^{k-1} - \mathbf{z} \right\|^2 - \frac{1 - d - w}{w} \left\| \mathbf{x}^{k-1} - T_w(\mathbf{x}^{k-1}) \right\|^2,$$
it follows that
$$\left\| \mathbf{x}^k - \mathbf{z} \right\|^2 = \left\| \mathbf{y}^k - \mu_k \nabla f(\mathbf{y}^k) - \mathbf{z} \right\|^2$$
$$= \left\| \mathbf{y}^k - \mathbf{z} \right\|^2 - 2\eta_k \langle \mathbf{y}^k - \mathbf{z}, \nabla f(\mathbf{y}^k) \rangle + \mu_k^2 \left\| \nabla f(\mathbf{y}^k) \right\|^2$$
$$\leq \left\| \mathbf{x}^{k-1} - \mathbf{z} \right\|^2 - \frac{1 - d - w}{w} \left\| \mathbf{x}^{k-1} - T_w(\mathbf{x}^{k-1}) \right\|^2$$
$$- 2\eta_k \langle \mathbf{y}^k - \mathbf{z}, \nabla f(\mathbf{y}^k) \rangle + \mu_k^2 \left\| \nabla f(\mathbf{y}^k) \right\|^2. \tag{16}$$

According to the known conditions, there exist $D_1, D_2 > 0$ such that
$$\left\| \mathbf{y}^k - \mathbf{z} \right\| \leq \left\| \mathbf{x}^{k-1} - \mathbf{z} \right\| \leq D_1, \left\| \nabla f(\mathbf{y}^k) \right\| \leq D_2.$$

If $u_k \leq 0$, then the inequality holds. Otherwise, applying (16) with $\mathbf{z}$ replaced by $\mathbf{x}'$, we obtain, for any $j \geq 1$,
$$\left\| \mathbf{x}^j - \mathbf{x}' \right\|^2 \leq \left\| \mathbf{x}^{j-1} - \mathbf{x}' \right\|^2 - 2\mu_j \langle \nabla f(\mathbf{y}^j), \mathbf{y}^j - \mathbf{x}' \rangle + \mu_j^2 D_2^2,$$
arrange the above inequality, we obtain
$$2\mu_j \langle \nabla f(\mathbf{y}^j), \mathbf{y}^j - \mathbf{x}' \rangle \leq \left\| \mathbf{x}^{j-1} - \mathbf{x}' \right\|^2 - \left\| \mathbf{x}^j - \mathbf{x}' \right\|^2 + \mu_j^2 D_2^2.$$

Since $\mu_k \geq \mu_j$ for all $k \leq j \leq 2k - 1$, summing it for all $j = k, k+1, \cdots, 2k-1$, yields
$$2 \sum_{j=k}^{2k-1} \mu_j \langle \nabla f(\mathbf{y}^j), \mathbf{y}^j - \mathbf{x}' \rangle \leq \left\| \mathbf{x}^{k-1} - \mathbf{x}' \right\|^2 - \left\| \mathbf{x}^{2k-1} - \mathbf{x}' \right\|^2 + \sum_{j=k}^{2k-1} \mu_j^2 D_2^2$$
$$\leq D_1^2 + D_2^2 \sum_{j=k}^{2k-1} \mu_j^2$$
$$\leq D_1^2 + D_2^2 (2k - 1 - k + 1)\eta_k^2$$
$$\leq D_1^2 + \frac{D_2^2 c^2 k}{(1 + k)^{2\alpha}}$$

By the definition of $u_k$ and $u_k \geq 0$, it follows that

$$2 \sum_{j=k}^{2k-1} \mu_j \langle \nabla f(\mathbf{y}^j), \mathbf{y}^j - \mathbf{x}' \rangle \geq 2u_k \sum_{j=k}^{2k-1} \mu_j \geq 2u_k k \eta_{2k-1} = c(2k)^{1-\alpha} u_k \geq ck^{1-\alpha} u_k,$$

then

$$u_k \leq \frac{D_1^2}{ck^{1-\alpha}} + \frac{cD_2^2 k^\alpha}{(1+2k+k^2)^\alpha} \leq \frac{D_1^2}{ck^{1-\alpha}} + \frac{cD_2^2}{k^\alpha}.$$

(ii) In order to prove the rate in terms of the outer objective function $f$, we will use (i) and for simplicity we define $\bar{k} = \arg\min\{\langle \nabla f(\mathbf{y}^j), \mathbf{y}^j - \mathbf{x}' \rangle : k \leq j \leq 2k\}$. We apply the sub-gradient inequality on the convex function $f$ to obtain

$$f(\mathbf{y}_{best}^k) - f(\mathbf{x}') \leq f(\mathbf{y}^{\bar{k}}) - f(\mathbf{x}') \leq \langle \nabla f(\mathbf{y}^{\bar{k}}), \mathbf{y}^{\bar{k}} - \mathbf{x}' \rangle \leq \frac{D_1^2}{ck^{1-\alpha}} + \frac{cD_2^2}{k^\alpha},$$

the first inequality follows from $f(\mathbf{y}_{best}^k) \leq f(\mathbf{y}^{\bar{k}})$, and the last inequality follows from (i). $\qquad\square$

# E  Other experiments about convergence

We evaluate the convergence of Algorithm 1 on the CelebA dataset for the Gaussian deblurring task with $\sigma_{blur} = 1, \sigma_{noise} = 0.02$. We set $K = 100, \mu_k = 9 \times 255\sigma_{noise}(k+1)^{-0.1}$. We first compute the spectral norm $\left\| J_{\nabla\psi_\theta}(\mathbf{x}^k) \right\|_*$, where $\{\mathbf{x}^k\}_{k=0}^K$ is the iterative sequence. Here we run the PI method with 200 iterations. As shown in Figure 6, we can estimate the tight Lipschitz constant $L_\theta = \max_{\mathbf{x}\in\mathbb{R}^n} \left\| J_{\nabla\psi_\theta}(\mathbf{x}) \right\|_* \geq 2.5$, then $2/L_\theta \leq 0.8, w \in (0, 0.8)$.

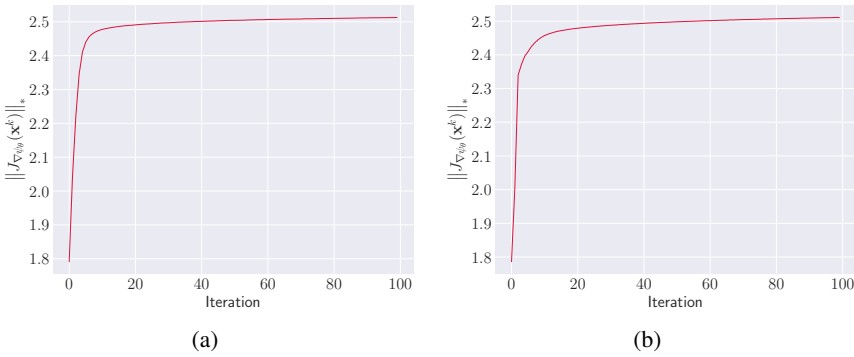

Figure 6: Spectral norm $\left\| J_{\nabla\psi_\theta}(\mathbf{x}^k) \right\|_*$ at the $k$-th iterative point $\mathbf{x}^k$ on two images.

We provide additional experiments to demonstrate the convergence of Algorithm 1 on four images in the CelebA dataset. The objective function and PSNR trends during iterations are shown in Figure 7.

# F  Hyperparameter Setting

We manually set the step size $\mu_k = \frac{c}{(1+k)^\alpha}$ and the weight $w$ of Algorithm 1 to achieve the best performance on the CelebA dataset. All hyperparameters are set to be the same for all images. The hyperparameters are summarized in Table 2.

Table 2: Parameter setting for CelebA dataset.

| Parameter | $\sigma_{blur} = 1, \sigma_{noise} = .02$ | $\sigma_{blur} = 1, \sigma_{noise} = .04$ | $\sigma_{blur} = 2, \sigma_{noise} = .02$ | $\sigma_{blur} = 1, \sigma_{noise} = .04$ |
|---|---|---|---|---|
| $c$ | $9 \times 255\sigma_{noise}$ | $9 \times 255\sigma_{noise}$ | $14 \times 255\sigma_{noise}$ | $20 \times 255\sigma_{noise}$ |
| $\alpha$ | 0.1 | 0.1 | 0.09 | 0.25 |
| $w$ | 0.5 | 0.75 | 0.75 | 0.75 |

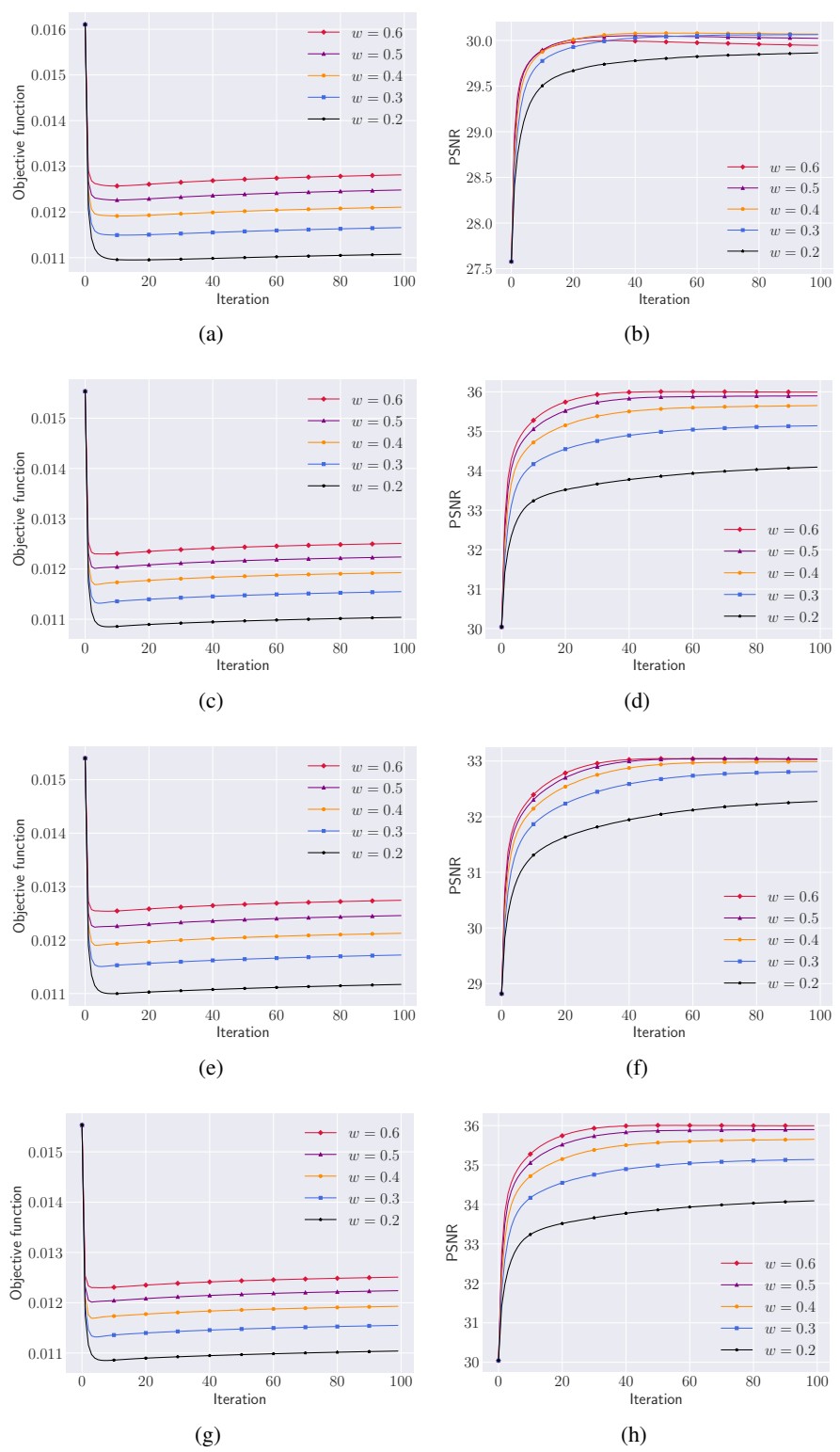

Figure 7: Convergence of Algorithm 1 on other images in the CelebA dataset. (a) Objective function. (b) PSNR.

# G Compared with state-of-the-art methods

We first train the SPC denoiser using the CelebA dataset and then apply Algorithm 1 to perform deblurring. We compare with diffusion-based method DiffPIR [69], PnP-PGD [34], and DPIR [66]. As demonstrated in Table 3, we evaluate the efficacy of RED-PRO with the learned truly SPC denoiser across a range of blur intensities, noise levels, and evaluation metrics. We give the hyperparameter setting of RED-PRO in Table 2, Appendix F. We provide visual comparisons in Figure 8. DPIR achieves the best results, our method ranks second, and DiffPIR effectively restores fine details but occasionally alters facial expressions.

Table 3: Deblurring results on CelebA over 20 samples.

| METHOD | $\sigma_{blur} = 1, \sigma_{noise} = .02$ | | $\sigma_{blur} = 1, \sigma_{noise} = .04$ | | $\sigma_{blur} = 2, \sigma_{noise} = .02$ | | $\sigma_{blur} = 2, \sigma_{noise} = .04$ | |
|---|---|---|---|---|---|---|---|---|
| | PSNR(↑) | SSIM(↑) | PSNR(↑) | SSIM(↑) | PSNR(↑) | SSIM(↑) | PSNR(↑) | SSIM(↑) |
| DiffPIR [69] | $30.8 \pm 2.0$ | $.86 \pm .03$ | $29.5 \pm 1.8$ | $.82 \pm .03$ | $28.6 \pm 2.0$ | $.80 \pm .05$ | $27.6 \pm 1.8$ | $.77 \pm .05$ |
| PnP-PGD [34] | $31.4 \pm 1.9$ | $.87 \pm .02$ | $27.6 \pm 0.9$ | $.71 \pm .05$ | $\underline{29.9 \pm 2.3}$ | $.85 \pm .05$ | $\underline{28.8 \pm 2.0}$ | $.81 \pm .05$ |
| DPIR [66] | $\mathbf{33.2 \pm 3.0}$ | $\mathbf{.92 \pm .03}$ | $\mathbf{31.8 \pm 2.6}$ | $\mathbf{.89 \pm .04}$ | $\mathbf{30.1 \pm 2.5}$ | $\mathbf{.86 \pm .05}$ | $\mathbf{29.1 \pm 2.2}$ | $\mathbf{.83 \pm .05}$ |
| RED-PRO | $\underline{32.4 \pm 2.8}$ | $.92 \pm .03$ | $30.8 \pm 2.3$ | $.88 \pm .03$ | $29.3 \pm 2.3$ | $.86 \pm .04$ | $28.4 \pm 2.0$ | $\underline{.83 \pm .04}$ |

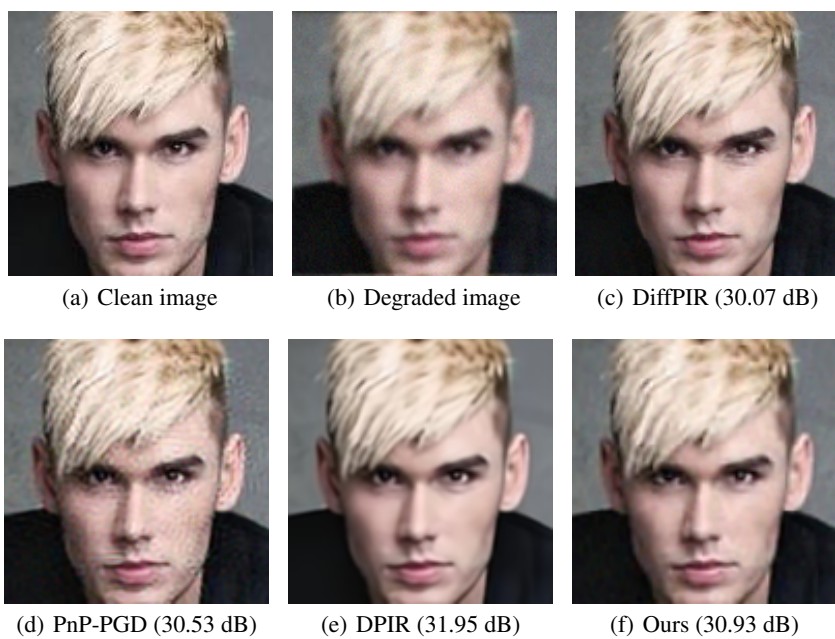

(a) Clean image      (b) Degraded image      (c) DiffPIR (30.07 dB)

(d) PnP-PGD (30.53 dB)      (e) DPIR (31.95 dB)      (f) Ours (30.93 dB)

Figure 8: Visual comparison on CelebA for Gaussian deblurring with $\sigma_{blur} = 1, \sigma_{noise} = 0.02$.

We consider a sparse-view computed tomography (CT) measurement model:

$$\min_{\mathbf{x} \in \text{Fix}(T_\theta)} \frac{1}{N} \sum_{i=1}^{N} \|\mathbf{A}_i \mathbf{x} - \mathbf{b}_i\|^2,$$

where $\mathbf{b}_i \in \mathbb{R}^m$ is the measured sinogram for the $i$-th projection, and $\mathbf{A}_i$ is an $m \times n$ discretized Radon transform matrix. For RED-PRO, we use $\mu_k = \frac{2}{\|\mathbf{A}\|^2 (1+k)^{0.01}}$, where $\mathbf{A} = [\mathbf{A}_1, \mathbf{A}_2, \ldots, \mathbf{A}_N]^{\mathrm{T}}$, and $w = 0.1$. The GS denoiser is trained on the public Mayo-CT dataset [**?** ]. We simulate CT sinograms using a parallel-beam geometry with 200 angles and 400 detectors. We compare with FBP and the recommended method in [38]. Table 4 presents the results for CT reconstruction. Despite RED-PRO's theoretical guarantees, its empirical performance in CT reconstruction may be inferior to that of PnP-ADMM.

Table 4: Numerical results for CT reconstruction on the Mayo-CT dataset, computed over 128 test images.

| Method | PSNR | SSIM |
|---|---|---|
| FBP | $20.233 \pm 0.034$ | $0.1763 \pm 0.0138$ |
| RED-PRO | $30.057 \pm 0.488$ | $0.8190 \pm 0.0075$ |
| PnP-ADMM [28] | $\mathbf{34.216 \pm 0.597}$ | $\mathbf{0.8938 \pm 0.0077}$ |

