# OpenReview forum: "Rethinking Gradient Step Denoiser: Towards Truly Pseudo-Contractive Operator"
_NeurIPS.cc/2025/Conference — NeurIPS 2025 poster_

### Official Review · Reviewer_Ygzw · 2025-06-06

**Clarity:** 3
**Significance:** 1
**Originality:** 2
**Rating:** 3
**Confidence:** 5

**Summary:**

This paper proposes a Gradient-Step denoiser with an ICNN as a learned potential in order to restore image with Plug-and-Play algorithm. The authors show that this Gradient-Step denoiser is a strictly pseudo-contractive operator, which allow to obtain convergence of the RED-PRO algorithm. This Gradient-Step denoiser does not require spectral norm penality in the training loss making the training of the denoiser more efficient.

**Questions:**

Major questions :
- The denoiser is train for one level of noise, can you propose strategy to train a denoiser with various level of noise, as DRUNet ?
- Why the Proposition 4.3 is restrict to convolutional neural networks ? I do not see where you use this convolutional structure in the proof.
- Why is it usefull to construct pseudo-contractive operators with Proposition 4.3 ? What is the difference with strictly pseudo-contractive operators in particular from a convergence point of view ?
- Do you observe instabilities when the constant $w$ violate your conditions ? Typically, $w > 0.8$ in Figure 5.

Minor questions:
- In the proof of Proposition 3.3, I do not see why the first and the second inequality are equivalent. Can you detail this computation ?
- In the proof of Proposition 3.3, you do not prove the reciprocal or exhibit that all arguments are equivalences.

**Ethical Concerns:**

["NO or VERY MINOR ethics concerns only"]

**Final Justification:**

I have raised my score to 3 to thank the authors efforts for answering my concerns.

I will still recommand to reject the paper because there is too many things to modify to the current version of the paper. The experiments are clearly unsufficient and it is hard to judge additionnal experiments only based on table (without looking closely to all the details). Moreover, as my review shows, I have detected numerous errors/imprecisions particularly in the references or in the statement of the theoretical results. Therefore I am not confident that I saw all major errors and that the results presented in the paper are correct.

**Limitations:**

The limitations of this work are not clearly exhibit. There is no comparison with state of the art methods in the experimental part. The question of expressivity of ICNN is not discuss in detail.

**Paper Formatting Concerns:**

- Before proposition 3.4, it is mention "\alpha$-averaged operators", is it "conically \alpha$-averaged operators", could you verify the consistency of your notation ?
- In Proposition 4.1, $\phi$ goes from $\mathbb{R}$ to $\mathbb{R}$
- Iterations (14) are not the same that iterations (13). (14) is the intermediar sequence of (13). Could you clearly shows that by using different notation for these two different sequences ?
- In line 220, you say that $T_w$ is $w \theta$ average, it must be $\frac{w}{1-d}$ to my understanding.
- In the proof of Proposition 4.6, you use $\eta_k$ instead of $\u_k$ for the steap-sizes.
- I suggests to recall what is $u_k$ in line 502
- In line 504, it might be $\eta_k \ge \eta_j$ instead of $\eta_k > \eta_j$
- In the proof of Theorem 4.6, the notation of the fix point change from $z$ to $x'$
- In line 256, I think that you mean MNIST, this error repeats.
- In line 262, the activation function is not softplus, there is an exponential missing.
- In Figure 2, I suggest to write "$T$ is $1-\frac{1}{\lambda}$-SPC" instead of "$T$ is $d$-SPC" to see clearly the link between $d$ and $\lambda$.

**Quality:**

1

**Strengths And Weaknesses:**

Strength:
The paper explains clearly and early the main point. The various operator conditions, FNE, SPC, PC, RFNE..., are well explained and their introduction is pedagogic (e.g. Figure 1). Reducing the time of training without spectral norm penality might interests practionners.

Weaknesses:
- There are numerous references mistakes and imprecisions:
	- For strictly pseudocontractive operator in the beggining of the introduction that have been studied for decades, you need to cite the first works such that [1,2]. With only the citation of [3], it makes the reader believe that these operators have been introduced last year.
	- For conically $\lambda$-averaged and nonexpansive, the references seems not very explored.
	- There is no precise references (which proposition or theorem ara copied) for books or article, especially for Proposition 3.2, 3.3, 3.4 or Theorem 3.6. Non precise reference makes it hard to look at.
	- There is no reference given for the different dataset.
	- In line 155, no reference is given for cocoercive operators.
	- In line 117, there is a wrong citation. DiffPIR is cited to support application not limited to image inverse problem. However, the method DiffPIR has been designed specifically for image inverse problem and is only applied to image inverse problem in the original paper that is cited.
	- In line 132, there is a missing citation. In fact "He et al." is cited, however in the references, no paper writing by He is provided.
- Some of the assumptions are not clearly stated. In Theorem 4.6, you suppose that the step-size are of the form $\mu_k = \frac{c}{(1+k)^\alpha}$ without saying it in the paper. It only appears in the proof in Appendix. This assumption must be clearly stated. In particular, it is need to write what is the condition on $\alpha$.
- The experimental validation is not strong. In line 91, it is claim that "extensive numerical experiments" are provided. However, only deblurring experiments are performed with few kernels of blur. Morevover, in Table 1, there is no comparison with state of the art PnP restoration method, such as using a DRUNet [4] or a Gradient-Step DRUNet denoiser [5], or the DiffPIR methode [6]. Then, RED-PRO do not outperform significatively PnP-DnCNN, restricting the impact of this work. Finally, no image are provided, even in Appendix, making it hard to have a quanlitative comparison between methods.
- There are imprecision in the proofs. In the proof of Proposition 3.2, in line 484, $d$ is suppose to be $-1$ where the Proposition claim that this is true for any $d < 1$. Moreover, according to Figure 2, Proposition 3.2 is not well formulate. In fact, it is not under the $d$-SPC that (i) is equivalent to (ii), but $d$-SPC is equivalent to (i) is and to (ii). I suggest to reformulate Proposition 3.2 to solve this error of logic.
- There is unprecise claims, which are false in the current version :
	- In line 74, you claim that gradient of ICNN possess universal approximation capabilities. However, the gradient of convex function are always monotone operators, so can not approximate every functions.
	- In equation (2), you claim that image restoration problems can be reformulate as a convex optimization problem. However, many image inverse problems have no-convex data-fidelity (e.g. phase retrieval) or non-convex regularization (e.g. Gradient-Step denoiser).

[1] On the Mann iteration process in a Hilbert space. Hicks, T.L., Kubicek, J.D., J. Math. Anal. Appl. 59, 498–504 (1977)
[2] Construction of fixed points of nonlinear mappings in Hilbert space, FE Browder, WV Petryshyn (1967)
[3] Andrzej Cegielski. Strict pseudocontractions and demicontractions, their properties, and applications. Numerical Algorithms, 95(4):1611–1642, Apr 2024.
[4] Plug-and-Play Image Restoration With Deep Denoiser Prior, Kai Zhang; Yawei Li; Wangmeng Zuo; Lei Zhang; Luc Van Gool; Radu Timofte
[5] Gradient step denoiser for convergent plug-and-play, Samuel Hurault, Arthur Leclaire, Nicolas Papadakis
[6] Denoising Diffusion Models for Plug-and-Play Image Restoration, Yuanzhi Zhu, Kai Zhang, Jingyun Liang, Jiezhang Cao, Bihan Wen, Radu Timofte, Luc Van Gool

---

> ### Author Rebuttal · Authors · 2025-07-30
>
> # To Reviewer Ygzw
> We sincerely appreciate your detailed and critical feedback. We have carefully addressed each of your concerns and made substantial improvements to the manuscript, including adding new experiments, clarifying theoretical contributions, and correcting technical details. We would greatly appreciate your reconsideration.
> ## Reply to Reference Corrections
> 1. Strictly Pseudocontractive Operators: We acknowledge that reference [2] you mentioned corresponds to our citation [8]. We will add reference [1] about Krasnosel'ski $\breve{ı}$-Mann (KM) iterations and restructure the introduction to present the earliest foundational works first, providing proper historical context.
> 2. Conically-averaged and nonexpansive References: We will add comprehensive citations about these operators, including:
> - Ref1. Bartz S, Dao M N, Phan H M. Conical averagedness and convergence analysis of fixed point algorithms[J]. Journal of Global Optimization, 2022, 82(2): 351-373
> - Ref2. Baillon, J.B., Bruck, R.E., Reich, S.: On the asymptotic behavior of nonexpansive mappings and semigroups in Banach spaces. Houston J. Math. 4, 1–9 (1978)
> - Ref3. Bauschke, H.H., Borwein, J.: On projection algorithms for solving convex feasibility problems. SIAM Rev. 38, 367–426 (1996)
> - Ref4. Combettes, P.L., Yamada, I.: Compositions and convex combinations of averaged nonexpansive operators. J. Math. Anal. Appl. 425(1), 55–70 (2015)
> - Ref5. Bauschke, H.H., Noll, D., Phan, H.M.: Linear and strong convergence of algorithms involving averaged
> nonexpansive operators. J. Math. Anal. Appl. 421(1), 1–20 (2015)
> - Ref6. Combettes, P.L.: Solving monotone inclusions via compositions of nonexpansive averaged operators. Optimization 53(5–6), 475–504 (2004)
> - Ref7. Combettes, P.L., Yamada, I.: Compositions and convex combinations of averaged nonexpansive operators. J. Math. Anal. Appl. 425(1), 55–70 (2015)
> 3. Precise Citations: Proposition 3.2 (i) follows from [Proposition 2.2, Ref1]. The equivalence of (i) and (ii) can be found in [14], page 1614, last sentence of the first paragraph. Proposition 3.3 is from [13, Corollary 2.2.3]. Proposition 3.4 is from [13, Corollary 2.2.3] and [56, Proposition 1 (b)]. Theorem 3.6 is based on [5, Theorem 5.8].
> 4. Dataset References: We add the following citations.
> - MNIST: LeCun Y, Bottou L, Bengio Y, et al. Gradient-based learning applied to document recognition[J]. Proceedings of the IEEE, 2002, 86(11): 2278-2324.
> - CelebA: Liu Z, Luo P, Wang X, et al. Deep learning face attributes in the wild[C]//Proceedings of the IEEE International Conference on Computer Vision. 2015: 3730-3738.
> - BSD400: D. Martin, C. Fowlkes, D. Tal, and J. Malik, “A database of human
> segmented natural images and its application to evaluating segmentation
> algorithms and measuring ecological statistics,” in Proceedings Eighth
> IEEE International Conference on Computer Vision. ICCV 2001, vol. 2,
> 2001, pp. 416–423 vol.2.
> 5. Specific Corrections:
> Line 155: The definition of cocoercive operator can be found in [5, Section 4.2, Definition 4.10].
> Line 117: We will correct the statement regarding DiffPIR.
> Line 132: We will add "He L, Zhang Q, Yang X, et al. SLN-RED:
> Regularization by simultaneous local and nonlocal denoising for image restoration[J]. IEEE Signal Processing Letters, 2023, 30: 578-582."
> ## Reply to assumptions on step-size
> We have mentioned the form of $\mu_k  =\frac{c}{(1+k)^\alpha}$ in Algorithm 1, not only in the Appendix. And $\alpha$ should satisfy that $\alpha>0$.
> ## Reply to experiments and limitations
> On the one hand, we provide a comparison with SOTA methods. Please refer to **Tables 4 and 5** at the end of the rebuttal to the 2nd **Reviewer Ln5k**. Although non-negative weights may limit ICNN expressivity, they ensure a balance between theoretical guarantees and empirical performance.
>
> On the other hand, our main contribution is theoretical: we take an important first step by showing how to construct truly SPC denoisers in Proposition 4.1, which provides new insights for designing interpretable and potentially more useful denoisers in the future. While our experimental validation is preliminary, we believe our work opens up new research directions and has the potential for significant impact.
> ## Reply to Proposition 3.2
> We set $d=-1$ as a special case to illustrate the equivalence between FNE and $1/2$-averaged operators. Then, we show that any nonexpansive operator $S$  can be written as a $2$-RFNE operator, i.e., $S= 2V-I$, where $V$ is FNE. By using the equivalence between conically averaged operators $T= (1-\lambda) I+\lambda S$ ($\lambda =\frac{1}{1-d}$) and $d$- SPC operators, by substituting $S = 2V - I$ we obtain $T = 2\lambda V+(1-2\lambda) I$, which can extend the result to any $d<1$. We will improve the exposition in the revised version. We will further clarify the logic of Proposition 3.2 by presenting the equivalence in parallel among $d$-SPC, conically $\lambda$-averaged, and $2\lambda$-RFNE, rather than the equivalence caused by the $d$-SPC condition, which may lead to misunderstanding.
> ## Reply to imprecise claims
> - Universal approximation capability:  We will revise the statement in line 74 to clarify that the universal approximation capability refers to monotone operators.
> - Model (2): We will revise the description of $f$ and $g$ in model (2) to cover more general inverse problems.
> ## Reply to Major questions:
> - **A1**: According to (7), it is possible to train a denoiser that handles multiple noise levels by randomly and uniformly sampling the noise level during training.
> - **A2 and A3**: The reason we mentioned CNNs is that [20] provides a method to control the Lipschitz constant of CNNs, so we presented a theoretical construction based on this, making general CNNs satisfy the pseudo-contractive assumption. ICNNs use non-negative weight constraints to ensure convexity, but this may reduce the expressive ability. We propose Proposition 4.3 to design new pseudo-contractive denoisers with stronger expressive power, but this is only a theoretical construction, and we plan to explore it in future work. Pseudo-contractive denoisers with $d=1$ have weaker conditions than strictly pseudo-contractive ones, but require the Ishikawa iteration to guarantee convergence, as shown in [51, Theorem 3.3]. Our theoretical analysis requires the denoiser to be  $d$-SPC.
> - **A4**: Yes, we do. Algorithm 1 requires $0 < w < 1-d$. Choosing a larger $w$ outside $(0,1-d)$ may violate the condition of Algorithm 1 and lead to instability in experiments.
> ## Reply to Minor questions
> - **A5**: Since
> $$\begin{aligned}
>  \|\| S(\mathbf{x}) - S(\mathbf{y}
> ) \|\|^2 & =\|\| (S(\mathbf{x})-\mathbf{x})+(\mathbf{x}-\mathbf{y})+(\mathbf{y}- S(\mathbf{y})) \|\|^2
> \\\\
> & = \|\|S(\mathbf{x})-\mathbf{x} \|\|^2+\|\| \mathbf{x}
> -\mathbf{y}\|\|^2+\|\| \mathbf{y}-S(\mathbf{y}) \|\|^2+2\langle S(\mathbf{x})-\mathbf{x},\mathbf{x}-\mathbf{y}\rangle \\\\
> &+2\langle S(\mathbf{x})-\mathbf{x},\mathbf{y}-S(\mathbf{y})\rangle+2\langle\mathbf{x}-\mathbf{y},\mathbf{y}-S(\mathbf{y})\rangle.
> \end{aligned}$$
> and $$
> \|\| (\mathbf{x}-S(\mathbf{x})) - (\mathbf{y}-S(\mathbf{y}))\|\|^2=\|\| \mathbf{x}-S(\mathbf{x}) \|\|^2-2\langle \mathbf{x}-S(\mathbf{x}), \mathbf{y}-S(\mathbf{y})\rangle+\|\| \mathbf{y}-S(\mathbf{y}) \|\|^2.$$
> From $$
>  \|\| S(\mathbf{x})-S(\mathbf{y}) \|\|^2\leq \|\| \mathbf{x}-\mathbf{y} \|\|^2 -\|\| (\mathbf{x}-S(\mathbf{x}))-(\mathbf{y}-S(\mathbf{y})) \|\|^2,$$
> we obtain
> $$\begin{aligned}
> \|\|\mathbf{x}-S(\mathbf{x}) \|\|^2+\|\| \mathbf{y}-S(\mathbf{y}) \|\|^2
> &\leq\langle \mathbf{x}-S(\mathbf{x}),\mathbf{y}-S(\mathbf{y})\rangle -\langle S(\mathbf{x})-\mathbf{x},\mathbf{x}-\mathbf{y}\rangle-\langle S(\mathbf{x})-\mathbf{x},\mathbf{y}-S(\mathbf{y})\rangle-\langle \mathbf{x}-\mathbf{y},\mathbf{y}-S(\mathbf{y})\rangle
> \\\\
> &=\langle S(\mathbf{y})-\mathbf{y}, S(\mathbf{x})-\mathbf{x}\rangle-\langle\mathbf{x}-\mathbf{y},S(\mathbf{x})-\mathbf{x}\rangle +\langle S(\mathbf{x})-\mathbf{x}, S(\mathbf{y})-\mathbf{y}\rangle+\langle\mathbf{x}-\mathbf{y},S(\mathbf{y})-\mathbf{y}\rangle
> \\\\
> &=\langle S(\mathbf{y})-\mathbf{x},S(\mathbf{x})-\mathbf{x}\rangle+\langle S(\mathbf{x})-\mathbf{y},S(\mathbf{y})-\mathbf{y}\rangle.\end{aligned}$$
> Finally, we show that these two inequalities are equivalent.
> - **A6**: We check all the derivations in the proof of Proposition 3.3 are equivalences. Please refer to [13, Corollary 2.2.3] or [14, Proposition 2.2] for a detailed justification. Proposition 3.3 is correct.
> ## Reply to Paper Formatting Concerns
> Thank you again for your careful and professional review. Some formatting concerns may be due to misunderstandings.
> - When $0<\alpha<1$, the term "$\alpha$-averaged" is commonly used in the literature[13,14,56, Ref1], rather than "conically $\alpha$-averaged", although $\alpha$-averaged is a special case of conically $\alpha$-averaged. Therefore, there is no issue of notation inconsistency in this regard.
> - We confirm that in Proposition 4.1, $\phi: \mathbb{R} \to \mathbb{R}$.
> - We respectfully disagree with the reviewer’s concern. Iterations (13) and (14) are equivalent, as clearly shown in the previous RED-PRO work published in SIIMS [18].
> - We will correct "$T_w$ is $w\theta$-averaged" to "$T_w$ is $\frac{w}{1-d}$-averaged".
> - We will use $\mu_k$ to replace $\eta_k$ in the proof of Theorem 4.6.
> - We will recall the definition of $u_k$ in line 502 for clarity.
> - We will correct the inequality in line 504 to "$\eta_k\geq \eta_j$".
> - We respectfully disagree. In our proof, $\mathbf{z}$ is introduced as an arbitrary fixed point and $\mathbf{x}'$ as the optimal solution. Replacing $\mathbf{z}$ with $\mathbf{x}'$ in the formula below line 502 is appropriate.
> - We will correct "MNIST" throughout the manuscript.
> - We will correct the activation function $\phi(x) = \frac{1}{\beta}\log\(1+\mathrm{e}^{\beta x}\)$.
> - We will update Figure 2 to use "$1-\frac{1}{\lambda}$ is-SPC" to clarify the relationship between $d$ and $\lambda$.

---

> ### Comment · Reviewer_Ygzw · 2025-08-05
>
> I thank the authors for these responses and details.
>
> I suggest to add all the corrected references in the paper.
>
> I suggest to add the stepsize definition in Theorem 4.6.
>
> Having proper theoretical guarantees for image restoration is crucial in many applications. However, if this is at the cost of a significant decrease of perfomances with respect with state-of-the-art methods, it is not useful. The experiments presented in the paper are clearly not sufficient to deduce the performances of the proposed methods. Concerning table 3-4-5, I truly appreciate the effort to provide additional experiments. I am surprised by the performance of DiffPIR, which denoiser have been used ? Also there is no comparison with the gradient-step denoiser, which is the closest method. Only "easily" inverse problem are tackle in these experiments, with low level of noise, low blur or low super-resolution factor. Therefore, it is hard to conclude the performance of your method. Finally, no images are provided (I know that it is not possible in the rebuttal format). I believe that a paper that is tackling the problem of image restoration should provide image for a qualitative comparisons.
>
> I thank the authors for the correction of Proposition 3.2.
>
> About typos disagreements :
> - I keep my comment, iteration (13) and (14) are not the same. If we name $y_k = x_k - \mu_k \nabla f(x_k)$ in equation (13), then $y_k$ is verifying (14).
> - Could you precise in line 502 that you take $z = x'$ for clarity ?

---

> > ### Author Response · Authors · 2025-08-06
> >
> > Thank you very much for your valuable suggestions. We will ensure that all corrected references, the stepsize definition in Theorem 4.6, and the correction of Proposition 3.2 are included in the revised manuscript. Regarding experiments, we have focused on demonstrating the convergence properties of RED-PRO and included comparisons with non-SPC methods, as well as reframed our experiments to highlight competitive results under theoretical constraints. For typos, we have carefully addressed all noted issues. We hope these comprehensive revisions address your concerns and demonstrate the value of our work.
> >
> > ## Regarding experiments
> > We appreciate your comments and agree that experimental validation is important. As suggested by Reviewer man6n, our revised experiments are designed to address "why use RED-PRO" by focusing on its convergence properties and providing clear cases where non-SPC methods fail to converge. In Table 6,  we further demonstrate that competitive results can be achieved while strictly satisfying theoretical constraints. Here we address your concerns about experiments:
> >
> > 1. The additional tables show that our method does not completely sacrifice performance; for some images, it achieves results comparable to state-of-the-art methods.
> > 2. For DiffPIR, we used its official codes based on the DDPM (Denoising Diffusion Probabilistic Model), employing a U-Net denoiser and the pre-trained model `diffusion_ffhq_10m.pth` as provided.
> > 3. In Tables 4 and 5, we used the official PnP-PGD code, which uses the gradient-step denoiser. Its lower performance in Table 4 may be due to a lack of pre-training on the CelebA dataset, while its good results in Table 5 reflect pre-training on a large natural image dataset.
> > 4. Regarding qualitative results, DPIR is widely recognized for both numerical and visual quality in this field. Our SSIM in Table 4 is comparable to DPIR, though our PSNR is lower, likely due to DRUNet's training on large datasets. We are confident that our method will also yield good visual results and will include qualitative comparisons in the final version.
> >
> > Overall, our experiments demonstrate that RED-PRO can achieve a balance between theoretical guarantees and practical performance.
> >
> > **Table 6**. Comparison of RED-PRO with various non-SPC denoisers on the Gaussian deblurring task. All non-SPC and the ICNN GS denoisers use Algorithm 1 with the same hyperparameters.
> >
> > | Methods|Parrot|House|Boat|Couple|Man|
> > |-|-|-|-|-|-|
> > |SPC-DnCNN [51]|25.73|31.43|28.53|28.17|29.58|
> > |MMO [38] |25.87|31.58|28.57|28.27|29.60|
> > |DnCNN [57]|25.30|26.74|26.23|25.98|26.47|
> > |DRUNet [56] |27.00|30.44|29.15|28.64|29.85|
> > |GS denoiser [28]|**27.38**|**32.58**|29.41|29.08|29.91|
> > |Ours |27.17| 32.40|**29.54**|**29.17**|**30.25**|
> >
> > ## Regarding typos
> > - We have revised equation (13) to $\mathbf{z}^k = T_{w}(\mathbf{z}^{k-1}-\mu_k \nabla f(\mathbf{z}^{k-1}))$;
> > - We agree and will clarify in line 502 that uses $\mathbf{z} = \mathbf{x}'$.

---

> ### Comment · Reviewer_Ygzw · 2025-08-07
>
> Thanks for this clear response. I will raise my score to 3 to thank the authors efforts for answering my concerns.
>
> I will still recommand a "weak reject" because there is too many things to modify to the current version of the paper (especially on experiments). Moreover, as my review shows I have detected numerous errors/imprecisions and it is highly possible that I missed others.
>
> Thanks for the discussion !

---

> > ### Author Response · Authors · 2025-08-09
> >
> > Thank you very much for your feedback and for raising your score in recognition of our efforts.
> >
> > We respectfully note that our work provides a novel theoretical contribution by proving that gradient step denoisers parameterized by ICNNs are inherently $d$-strictly pseudo-contractive, which advances the understanding of PnP/RED methods. We believe our work has the potential for significant impact.
> >
> > We appreciate your comments, which are valuable for further improving our work. However, we believe the significance of our contribution should not be judged solely on partial experimental results or minor formatting issues. We have made substantial efforts to address your main concerns, particularly in experiments. Thank you again for your thoughtful discussion.

---

### Official Review · Reviewer_ma6n · 2025-06-20

**Clarity:** 1
**Significance:** 2
**Originality:** 2
**Rating:** 4
**Confidence:** 4

**Summary:**

This paper considers the problem of learning a denoiser which possesses the pseudo-contractive property, for use in solving inverse problems via the plug-and-play (PnP) and regularization-by-denoising (RED) techniques. This property (or special cases of it) on a denoiser appears in various convergence analyses of PnP and RED methods. Learning denoisers with this property is a challenging task, and the paper describes the limitations of spectral methods that have been proposed to solve this. The key contribution of this paper is that a gradient step (GS) denoiser, using the gradient of an $L_\theta$-smooth input-convex neural network (ICNN), is a strictly pseudo-contractive (SPC) operator with parameter depending on $L_\theta$. It also shows that a Lipschitz continuous neural network is pseudo-contractive and presents a quantitative convergence result for RED-PRO with these gradient step denoisers. Experiments are performed: (1) testing the pseudo-contractive property of two denoisers trained using the spectral method and one ICNN GS denoiser, finding that the spectral method denoisers violate the SPC property whilst the GS denoiser does not; (2) exploring the convergence of RED-PRO with an ICNN GS denoiser; and (3) testing RED-PRO with an ICNN GS denoiser vs. two PnP methods for deblurring CelebA images.

**Questions:**

* How does the theoretical analysis of PnP/RED methods with pseudo-contractive denoisers depend on $d$? Crucially, do the values of $d = (L_\theta-2)/L_\theta$ and $d = 1$ in Propositions 4.1 and 4.3 significantly affect the ability of these Propositions in establishing any meaningful guarantees for the resulting PnP/RED method(s)?
* For the ICNN GS denoiser, does the $L_\theta$-smoothness need to be carefully controlled (for example, if Proposition 4.1 needs to guarantee that the ICNN GS denoiser must be non-expansive or even FNE)—and if so, how practical is this?—or can $L_\theta$ be as large as you like whilst still achieving good guarantees?
* For Proposition 4.3, why restrict $R_\theta$ to be a CNN and, much more importantly, when is $T_\theta$ meaningfully a “denoiser”? For example, if $R_\theta = 0$, then $T_\theta = (1-L)I$ for any choice of $L \geq 0$, which isn’t really a denoiser. If $R_\theta$ is bounded but varies quickly, and hence has Lipschitz constant much higher than its bound, then $T_\theta$ will just be a small perturbation of $(1-L_\theta)I$, and so also be no use at denoising; that [20] only gives an upper bound on $L_\theta$ will amplify this issue. What properties must $R_\theta$ have for $T_\theta$ to be a useful denoiser?
* Given the size of the errors in the PSNR and SSIM relative to the advantage of RED-PRO vs. PnP-DnCNN, does RED-PRO really “outperform” PnP-DnCNN for any parameters? What does a qualitative comparison of the deblurred image reveal? Although I appreciate that this is theoretical work and not trying to  achieve state-of-the-art, PnP-DnCNN is 8 years old; can you explain why one should expect practical benefits from using RED-PRO with an ICNN GS denoiser?
* Two confusions/typos. Line 262: Isn’t softplus $\frac{1}{\beta} \log (1 + e^{\beta x})$? Is this a typo or did you use a different activation to softplus? Line 272: Should this say “exceeds 1” not “exceeds -1”?

**Ethical Concerns:**

["NO or VERY MINOR ethics concerns only"]

**Final Justification:**

The authors have addressed the bulk of my concerns in their rebuttals. I raise my score to 4, but no higher, because without reading the revised manuscript in full it is impossible to tell if my clarity concerns have been fully addressed.

**Limitations:**

* Unclear whether the $d$ values for pseudo-contractivity guaranteed by the key results produce useful guarantees from the theory. If they do, authors should make this clear. If they do not, authors should discuss, e.g., how the $L$-smoothness can be controlled to ensure suitable $d$ values.
* Unclear under which conditions Proposition 4.3 yields a useful denoiser.
* Experimental deblurring results do not show any significant advantage.

**Paper Formatting Concerns:**

I have no formatting concerns.

**Quality:**

2

**Strengths And Weaknesses:**

# Quality

## Strengths

* Key result is of potential practical interest.
* Theorems all appear to be correct, with valid, straightforward proofs.
* Related work is adequately described.

## Weaknesses

* The key result (Proposition 4.1) is a fairly simple consequence of known results.
* Their alternative construction (Proposition 4.3) does not indicate how to learn a pseudo-contractive denoiser: the obtained operator T may not do anything resembling denoising.
* The link to the core motivation is not established: the existing theory for PnP/RED which supposedly uses these properties is not clearly stated, making it hard to judge the value of the properties established by these propositions.
* Estimating and controlling the L-smoothness/Lipschitz parameter may be important for making use of these results, yet is mentioned only off-hand.
* The inverse problem considered is not especially complex, and the results show no significant improvement for RED-PRO (any gains over PnP are much smaller than the errors). No example outputs are given, preventing qualitative assessment.
* Authors do not discuss limitations.

# Clarity

## Strengths

* Proofs are clearly written.
* Figures/tables are easy to interpret.
* Contributions are clear.

## Weaknesses

* Far too many technical details are frontloaded in the introduction, obscuring the story of the work, and technical definitions are dispersed between introduction and Preliminaries. Clarity would be greatly enhanced by focusing paragraph 1 on the story of this work: what the problem is, why it matters, and how this work answers it, rather than immediately throwing a lot of technical terms and definitions at the reader, which can be moved to Preliminaries.
* Technical details of spectral methods don’t belong in the introduction, move to Related Works. Focus on the ideas, advantages, and disadvantages of those methods in the introduction.
* “The pseudo contractive assumption has played an important role in the convergence analysis of PnP methods [50, 42, 10, 40, 46, 49, 31, 36, 47, 37, 24, 3, 7, 51, 35, 41, 6]  and RED framework [39, 38, 18, 33]” some of these references do not appear to be  relevant to this claim. Clarify this.
* Given the centrality of the claim to the work, it should be crystal clear why pseudo-contractivity is important for PnP and RED methods. Include in Related Work any theory which you will later rely on to motivate results in the Preliminaries and/or Section 4, so that it is very clear what the properties established in Propositions 4.1 and 4.3 entail. For example, it is unclear why Proposition 3.3 is a key motivation for anything.
* Some awkward grammar/typos, e.g., lines 77, 200, 205, and 291.

# Significance

## Strengths

* The key results show a straightforward way to guarantee a property related to desired properties in practice.

## Weaknesses

* As mentioned above, the authors could do more to clarify the significance of these results.
* No significant advantage was observed in inverse problem experiments

# Originality

## Strengths

* To my knowledge of the literature, the key results are novel

## Weaknesses

* The key results are largely simple consequences of known results.

---

> ### Author Rebuttal · Authors · 2025-07-30
>
> # To Reviewer ma6n
> In summary, we believe our work provides valuable new theoretical insights and establishes a principled framework for designing interpretable denoisers with provable guarantees. We sincerely hope that our clarifications and new experimental results highlight the theoretical and practical significance of our contributions and effectively address your concerns. And we would greatly appreciate your reconsideration.
> ## Reply to Weaknesses of Quality
> - **A1:** Although Proposition 4.1 is a direct consequence of established results, most existing theoretical analyses of PnP/RED methods, to our knowledge, depend on restrictive assumptions regarding the denoiser, such as nonexpansiveness or averaged nonexpansiveness, which are often not satisfied in practice. Our work is the first to reveal the connection between GS denoisers and truly pseudo-contractive denoisers. This connection enables us to obtain truly SPC denoisers without relying on spectral methods, which is crucial for the theoretical analysis of PnP/RED methods. Moreover, Theorem 4.6 provides the convergence rate of the objective function for the RED-PRO method, which is a new result in this field.
> - **A2:** Proposition 4.3 presents a theoretical framework for constructing pseudo-contractive denoisers. The condition required for $R_\theta$ is $L_\theta$-Lipschitz continuity. In practice, a meaningful $\tilde{R}_ {\theta}$ needs to be learned from data. The learning process involves minimizing the loss function:
>
> $$
> \min_{\theta} \{ \mathcal{L}(\theta) = \mathbb{E}_{\mathbf{x}\sim p(\mathbf{x}), \mathbf{n}\sim \mathcal{N}(\mathbf{0},\sigma^2\mathbf{I})} || T _{\theta} (\mathbf{x}+\mathbf{n})-\mathbf{x} ||^2 \},
> $$
>
> where $T_{\theta}  = I-\tilde{R}_ {\theta}, \tilde{R}_ {\theta} =R_ {\theta}+L_{\theta} I $.
>
> Reference [20] provides an effective method to estimate the Lipschitz constant of a network, which can be combined with Proposition 4.3 to learn pseudo-contractive denoisers effectively.
> - **A3:** The existing theory for some PnP/RED methods relies on assumptions such as nonexpansive, FNE, or averaged nonexpansive operators, but it is actually difficult to obtain denoisers that strictly satisfy these mathematical properties in practice. For example, [37, Assumption 2] requires the residual $R_\sigma = I - D_\sigma$ to be FNE, so is $D_\sigma$; [42, Theorem III.1] requires the denoiser to be a proximal mapping for some proper closed convex function, which is also FNE; [38, Lemma 5] requires the denoiser to be nonexpansive; [36, Theorem 3.5, Theorem 3.6] and [46, Assumption 2(b)] require the denoiser to be $\theta$-averaged with $\theta \in (0,1)$; [33, Assumption 2] requires the denoiser to be contractive. All these definitions fall under the category of pseudo-contractive operators. One of our core contributions is Proposition 4.1, which proves that GS denoisers based on ICNNs are truly SPC denoisers. We will clarify in future revisions how Proposition 4.1 has practical significance in the theory of PnP/RED methods.
> - **A4:** Figure 5 illustrates the use of the power method for estimating the global lower bound of $L_\theta$, which can assist in tuning $w$. Reference [20] provides a method for estimating the Lipschitz constant of a network, but how to precisely and efficiently control the Lipschitz constant of neural networks remains an open problem. We will continue to investigate this problem in our future work.
> - **A5:** Thank you for your comment. Our work is mainly focused on theoretical analysis and the connection between GS denoisers and pseudo-contractive operators, rather than on proposing a new state-of-the-art method for complex inverse problems. We have added CT reconstruction and super-resolution experiments in **Table 2** (see rebuttal to Reviewer 1SLa) and **Table 3** (see rebuttal to Reviewer Ln5k), respectively. Additionally, **Tables 4 and 5** (see rebuttal to Reviewer Ln5k) provide comparison experiments with state-of-the-art methods. We acknowledge that pursuing interpretability may sacrifice some performance, and we will consider how to better balance the two aspects in future work. Due to this year's conference rebuttal rules, we will provide visual results in the revised manuscript.
> - **A6**：We discussed some limitations briefly in lines 199–201, highlighting the trade-off between performance and interpretability. In the revised manuscript, we will provide a more detailed analysis.
>
> ## Reply to Weaknesses of Clarity
> - **A7:** Thank you for your suggestion. Our motivation starts from the drawbacks of spectral methods. In the revised manuscript, we will consider moving the technical details of spectral methods to the Related Work Section to improve readability. Additionally, we will clarify how the PnP/RED theory relies on the mathematical properties of pseudo-contractivity, further highlighting the value of our work.
> - **A8**: We explained in lines 26–28 which operator assumptions are used in these references, as they are relevant to this field. We also provide some specific examples in **A2** illustrating how these mathematical properties are used in related works.
> - **A9**: We believe there may be some misunderstanding regarding our motivation. The starting point of our work is that spectral methods cannot yield truly SPC denoisers, while existing PnP and RED theories rely on this property for theoretical guarantees. Proposition 3.3 is important because it helps us reveal the connection between GS denoisers and truly SPC denoisers. In the revised manuscript, we will clarify the role of Proposition 3.3, rather than describing it as a "key motivation" to avoid any misunderstanding.
> - **A10**: Thank you. We will carefully proofread and correct the issues in lines 77, 200, 205, 291, and throughout the manuscript.
>
> ## Reply to Weaknesses of Originality
> - **A11**：First, $d$ only affects parameter selection, specifically $w \in (0, 1-d) = (0, \frac{2}{L_\theta})$ (see line 226). Choose $w \in (0, 1-d)$ appropriately so that $ T_w = wT_ \theta + (1-w)I $ is a nonexpansive operator. It does not affect the main theoretical results, such as the convergence rate in Theorem 4.6. Theoretical analysis of PnP/RED methods relies on $d$-SPC assumptions such as nonexpansive ($d=0$), FNE ($d=-1$), and $\alpha$-averaged nonexpansive ($d=\frac{\alpha-1}{\alpha}$) operators. Therefore, achieving the truly pseudo-contractivity property is important for the theoretical understanding of PnP/RED methods. Our theoretical results, i.e., Theorem 4.6,  are mainly based on $d$-SPC denoisers. For the case $d=1$, Proposition 4.3 provides only a theoretical construction of pseudo-contractive denoisers.  Ishikawa iteration presented in reference [51, Theorem 3] can be used to ensure convergence of PnP methods.
> - **A12**: In our work, we don't precisely control the smoothness constant $L_\theta$. Theoretically, $L_\theta = 2$ corresponds to a nonexpansive operator and $L_\theta = 1$ to a FNE operator. However, how to precisely control the network's Lipschitz constant $L_\theta$ remains a very important open problem; in our experiment shown in Figure 4, we only provide a rough estimate. As far as we know, reference [20] provides a recent method to estimate an upper bound for the network's Lipschitz constant, which can help prevent $L_\theta$ from becoming too large. Achieving precise control of $L_\theta$ to obtain desired operators such as FNE or nonexpansive is still challenging and worth further exploration in the future. In our experiments, we mainly tune the weight $w$ to ensure that $T_w$ remains non-expansive. Since $w \in (0, \frac{2}{L_\theta})$, a larger $L_\theta$ requires a smaller $w$ to ensure good theoretical guarantees.
> - **A13**: Because [20] provides a way to control the Lipschitz constant of CNNs, which are widely used in image processing. Theoretically, we aim to design new pseudo-contractive denoisers. When $R_\theta$ is trained in a data-driven manner, $T_\theta$ will be a meaningful denoiser. For further explanation, please see our response to **A2**.
> - **A14**: No, RED-PRO does not always outperform PnP-DnCNN. We have tuned $w$ and step size $\mu_k$ to achieve better performance than PnP-DnCNN in some cases. We will provide some qualitative results to observe their difference in the revised paper. The practical benefit of using RED-PRO with an ICNN GS denoiser is that the ICNN GS denoiser is guaranteed to satisfy the strict pseudo-contractivity (SPC) property, which ensures the theoretical convergence of RED-PRO. This provides a principled way to design denoisers that are both interpretable and compatible with the convergence guarantees of RED-PRO.
> - **A15**: We will correct two confusions/typos about the softplus function and the expression “exceeds -1”.
> ## Reply to limitations
> - **A16**: For Proposition 4.1, we have $d = \frac{L_\theta-2}{L_\theta}$, and we only need to tune the weight $w \in (0,1-d) =  (0, \frac{2}{L_\theta})$ in Algorithm 1. The experiments in Figure 5 show that the obtained $d$ values can indeed provide convergence guarantees. However, as $L_\theta$ increases, $w$ must be smaller. In future work, we will explore how to obtain better $d$ values by accurately controlling the network's $L$-smoothness.
> - **A17**: Proposition 4.3 provides a theoretical condition by requiring the Lipschitz constant of the network to be controlled in advance. To obtain a useful denoiser in practice, it is necessary to train the denoiser in a data-driven manner.
> - **A18**: We acknowledge that the experimental deblurring results do not show a significant advantage. Our main contribution is to provide new theoretical insights. Additional experiments, such as those in **Tables 4 and 5** (see rebuttal to the **Reviewer Ln5k**), show that RED-PRO with truly SPC denoisers can achieve a balance between theoretical guarantees and practical performance compared to state-of-the-art methods.

---

> ### Comment · Reviewer_ma6n · 2025-08-04
>
> I thank the authors for their thorough response to my review. I will here follow up on where my most significant concerns remain.
>
> * **A2/A4/A13**: This point deserves to be fleshed out more in the paper. Proposition 4.3, as written, tells us how to turn an L-Lipschitz NN into a pseudo-contractive operator, but this need not be a denoiser. I thank you for clarifying in your reply that you intend to get a denoiser by training the resulting operator via a denoising loss, but this training will require considering the Lipschitz constant of the underlying $R_\theta$ neural network during training. The discussion of how to get a pseudo-contractive **denoiser** out of Proposition 4.3 would therefore be greatly enhanced by a discussion of training Lipschitz-constrained neural networks in this setting, going beyond the simple reference to [20]. For example, there are known challenges regarding expressivity, see "Sorting Out Lipschitz Function Approximation" by Cem Anil, James Lucas, Roger Grosse  (2019) and "Limitations of the Lipschitz constant as a defense against adversarial examples" by Todd Huster, Cho-Yu Jason Chiang, Ritu Chadha (2018).
> * **A3/A7/A11/A12**: The heart of my issue is this: Proposition 4.1 shows that the ICNN GS denoiser will be SPC with $d = (L_\theta-2)/L_\theta$. But as you say in your reply, the existing theory for PnP/RED relies on properties that are *more specific* than simply being SPC--being SPC is a necessary but not a sufficient condition for this theory. As you note, this necessary condition is not achieved by other methods, but the way things are currently being presented in your paper, it is highly unclear to the reader what theory the GS denoiser gains automatically from being $d$-SPC (like Theorem 4.6) vs. what will require careful control of $L_\theta$ (e.g., if FNE were required, then one would need $L_\theta \leq 1$). Tidying this up (and adding discussion/references for how $L_\theta$ might be controlled, even if actually doing so is beyond the scope of this work).
> * **A14**: My concerns are two-fold: Firstly, you should not describe RED-PRO as outperforming another method if the difference in performance is much smaller than the error bars, as this miscommunicates the (statistical) significance of the performance. (Ideally, one would perform a significance test of RED-PRO vs. PnP-DnCNN/etc., and report it as outperforming only when results were statistically significantly better.) In the results in the paper, and from the above Tables, it seems that RED-PRO rarely if ever has significant improved performance, and this should be communicated clearly. Secondly, therefore, ideally your experiments should seek to highlight "why should I use RED-PRO?" in some other way--perhaps focusing on the convergence property, showcasing cases where the non-SPC methods fail to converge. Alternatively, since as you say state-of-the-art experimental performance are not this work's focus, you could reframe your experiments as showcasing how competitive results can be attained whilst still satisfying the constraints of theory.
>
> If these concerns are met, I will consider raising my score to a 4. Please reply with excerpts of what you add with respect to these concerns, so I can get a sense of what the revised manuscript would look like.

---

> > ### Author Response · Authors · 2025-08-06
> >
> > Thank you for your thoughtful follow-up and constructive suggestions. Reference numbers correspond to those in the revised paper. Please see the following specific excerpts.
> > ## Theoretical benefits
> > Our Proposition 4.1 is the first to realize a truly $\frac{L_\theta-2}{L_\theta}$-SPC operator via the ICNN GS denoiser, whose assumption is weaker than FNE and nonexpansive, and thus easier to satisfy in practice. Once the GS denoiser meets the $\frac{L_\theta-2}{L_\theta}$-SPC condition, the RED-PRO framework automatically guarantees sequence convergence and objective convergence rate, as shown in Theorems 4.5 and 4.6, without requiring stronger FNE or nonexpansive assumptions. The result of Proposition 4.1 can further benefit existing PnP/RED theoretical works by enabling the ICNN GS denoiser to satisfy stronger FNE or averaged nonexpansive assumptions in two ways:
> >
> > - Controlling $L_\theta \leq 1$, e.g., by normalizing convolutional kernels via spectral methods [41], orthonormalization [3], and penalizing the network's Lipschitz constant in the loss [29], so that the FNE and nonexpansive assumptions required in [47, Assumption 2], [43, Theorem III.1], and [39, Lemma 5] are met;
> > - Estimating $L_\theta$ via the power method and tuning the weight $w<\frac{2}{L_\theta}$ so that $T_w=wT_\theta+(1-w)I$ is a $\frac{wL_\theta}{2}$-averaged operator, thus satisfying the averaged operator assumptions in [46, Assumption 2(b)] and [37, Theorem 3.5, Theorem 3.6].
> >
> > Therefore, Proposition 4.1 provides valuable practical guidance for existing PnP/RED theoretical works that require stronger FNE and averaged nonexpansive assumptions.
> >
> > ## Discussion of pseudo-contractive denoiser
> > The core problem in Proposition 4.3 is how to train Lipschitz-constrained neural networks. Ryu et al. normalized each convolutional kernel to obtain a $1$-Lipschitz CNN. Anil et al. [3] proposed that by combining GroupSort activation functions with orthonormal weight matrices, one can construct networks that are provably $1$-Lipschitz and capable of approximating any $1$-Lipschitz function arbitrarily well. These methods can be used to train the $1$-Lipschitz-constrained neural networks $R_ \theta$ in Proposition 4.3. In this case, the Lipschitz constant $L_\theta$ is equal to $1$, then $\tilde{R}_ \theta = R_\theta + I$, and $\tilde{R}_ \theta=R_\theta+I$ can be viewed as a residual connection, which is used to fit the noise distribution $\mathbf{n}$. That is, $\tilde{R}_ \theta$ is obtained by minimizing the following loss function:
> > $$
> >   \min_{\theta} \\{\mathcal{L}(\theta)=\mathbb{E}_ {\mathbf{x}\sim p(\mathbf{x}),\mathbf{n}\sim\mathcal{N}(\mathbf{0},\sigma^2\mathbf{I})}\|\|\tilde{R}_ \theta(\mathbf{x}+\mathbf{n})-\mathbf{n}\|\|^2\\},
> > $$
> > and the pseudo-contractive denoiser $T_\theta=I-\tilde{R}_ \theta$ is constructed. Moreover, Delattre et al. [20] controlled a $L$-Lipschitz convolutional kernel $\mathcal{K}_ j(1\leq j\leq l)$, the training loss becomes: $\mathcal{L}(\theta)+\mu_ {\mathrm{reg}}\sum_ {j=1}^l \mathcal{L}_ {\mathrm{reg}}(\mathcal{K}_ j)$ with
> > $$
> > \mathcal{L}_ {\mathrm{reg}}(\mathcal{K}_ j)=\sigma_{\mathrm{GI}}(\mathcal{K}_ j) \mathbf{1}_ {\sigma_{\mathrm{GI}}(\mathcal{K}_ j)>L},
> > $$
> > where $\sigma_{\mathrm{GI}}$ denotes the spectral norm computed by Gram iteration (GI), which is more efficient and accurate than PI, and $x\to\mathbf{1}_{x>L}$ indicates 1 if $x>L$, and 0 otherwise.
> >
> > Although the above methods theoretically yield pseudo-contractive denoisers by training Lipschitz-constrained neural networks, such networks may suffer from limited expressive capacity [3,30]. For example, Anil et al. proved that 2-norm-constrained networks with ReLU (or sigmoid, tanh, etc.) activations cannot represent the absolute value function [3], and Huster et al. pointed out that Lipschitz-based approaches suffer from representational limitations that may hinder performance on complex tasks [30].
> >
> > ## Competitive results with theoretical guarantees
> > We have removed statements such as "RED-PRO outperforms PnP-DnCNN" and replaced “Application to inverse problems” with “Competitive results with theoretical guarantees.” Following your suggestion, we have reframed the experiments in Table 6 to highlight how competitive results can be achieved while strictly satisfying theoretical constraints. As in Appendix G.5 of Hurault's work [29], we will also provide visual convergence curves to clearly demonstrate that non-SPC methods may not converge.
> >
> > **Table 6**. Comparison of RED-PRO with various non-SPC denoisers on the Gaussian deblurring task. All non-SPC and the ICNN GS denoisers use Algorithm 1 with the same hyperparameters.
> >
> > | Methods|Parrot|House|Boat|Couple|Man|
> > |-|-|-|-|-|-|
> > |SPC-DnCNN [51]|25.73|31.43|28.53|28.17|29.58|
> > |MMO [38] |25.87|31.58|28.57|28.27|29.60|
> > |DnCNN [57]|25.30|26.74|26.23|25.98|26.47|
> > |DRUNet [56] |27.00|30.44|29.15|28.64|29.85|
> > |GS denoiser [28]|**27.38**|**32.58**|29.41|29.08|29.91|
> > |Ours |27.17| 32.40|**29.54**|**29.17**|**30.25**|

---

> > > ### Comment · Reviewer_ma6n · 2025-08-06
> > >
> > > I thank the authors for their substantial efforts to address my concerns. I am now satisfied that this paper is suitable for acceptance, and will adjust my score.
> > >
> > > As an unrelated suggestion, in your response to 1SLa, you include a CT experiment, but compare only to FBP. FBP is a bit too easy to beat in my experience, and for the camera-ready I would suggest including comparisons to other methods, for example a subset of the methods considered in "Benchmarking Learned Algorithms for Computed Tomography Image Reconstruction Tasks" by Kiss et al. (2025).

---

> > > > ### Author Response · Authors · 2025-08-09
> > > >
> > > > Thank you very much for your positive feedback and helpful suggestion. As promised, we will include comparisons to your recommended method in the final version, i.e., "Benchmarking Learned Algorithms for Computed Tomography Image Reconstruction Tasks" by Kiss et al. (2025).

---

### Official Review · Reviewer_Ln5k · 2025-06-23

**Clarity:** 4
**Significance:** 3
**Originality:** 2
**Rating:** 5
**Confidence:** 5

**Summary:**

The RED-PRO algorithm offers a robust approach to solving inverse problems by incorporating a denoiser within a fixed-point iteration, with convergence proven for the broad family of demicontractive operators, in particular pseudo-contractive denoisers. However, the practical construction of pseudo-contractive denoisers has presented a significant open challenge. This paper directly addresses this by adopting gradient-driven, or gradient-step, denoisers. The main contribution is a theoretical proof establishing that these gradient-driven denoisers are indeed pseudo-contractive, thereby guaranteeing RED-PRO's convergence when they are employed. Furthermore, the authors present an additional theoretical result detailing how to construct a pseudo-contractive denoiser from a Lipschitz network. Experimental validation in image deblurring supports these theoretical claims, confirming the pseudo-contractive nature of the trained denoisers, the convergence of RED-PRO, and showcasing their favorable performance in PSNR and SSIM.

**Questions:**

1.  Please clarify in the experimental section whether the denoiser used is gradient-based or constructed using Proposition 4.3.
2. Elaborate on the advantages and disadvantages of gradient-step denoisers compared to denoisers built upon Proposition 4.3.
3. Extend the experiments to include other inverse problems, such as super-resolution, and an application where convergence, stability, or interpretability is crucial (e.g., medical imaging).

**Ethical Concerns:**

["NO or VERY MINOR ethics concerns only"]

**Limitations:**

While the authors prioritize theoretical interpretability over state-of-the-art performance, the paper should still discuss the limitations and applicability of gradient-step denoisers within the RED-PRO algorithm. This discussion should compare them against other state-of-the-art approaches in both academic and real-world settings, detailing their respective pros and cons, performance gaps, and potential future strategies for improvement.

**Paper Formatting Concerns:**

No formatting issues.

**Quality:**

4

**Strengths And Weaknesses:**

Strengths:
This paper is well-written, clearly presenting the problem it addresses along with the necessary background and relevant literature. The theoretical derivations, which form the core contribution, are clear and well-supported. The significance of this theoretical work lies in the fact that it directly leads to a practical algorithm for inverse problems that offers provable stability and convergence, a characteristic that can be crucial in many real-world applications.

Weaknesses:
1. The paper's theoretical strength lies in its straightforward connection of two prior works to achieve provable convergence. However, this simplicity simultaneously suggests a limited degree of independent originality.
2. The current experiments, limited to image deblurring, could be expanded. Extending them to other well-known inverse problems would significantly strengthen the case for RED-PRO with SPC denoisers as a general solution.
3. Continuing the previous point, in common academic inverse problems like image super-resolution, approaches such as diffusion models often exhibit significantly superior performance to RED-PRO. Therefore, I recommend including experiments in applications where stability is crucial, thereby highlighting RED-PRO's relevance and utility in practical scenarios.
4. Proposition 4.3, though linked to the paper's broader discussion, does not seem to fit seamlessly within the focus of the paper, which can slightly obscure the main contribution.

---

> ### Author Rebuttal · Authors · 2025-07-30
>
> # To Reviewer Ln5k
> We sincerely appreciate your constructive and positive feedback.  We have addressed your main concerns regarding theoretical originality, experimental validation, and practical relevance as detailed below.
> ## Reply to Weaknesses
> - **Theoretical Originality:** We acknowledge that our theoretical contribution primarily establishes a straightforward connection to prior works, and we appreciate your feedback regarding the level of originality. Existing theoretical studies on PnP/RED methods typically assume that the denoiser is either FNE, nonexpansive, or averaged nonexpansive, all of which are special cases of pseudo-contractive denoisers. While most studies employ spectral methods to construct denoisers intended to satisfy these assumptions, but in practice, they often do not meet the theoretical conditions. Our work is the first to provide a theoretical proof that the gradient-step denoiser corresponds to a pseudo-contractive operator, which is crucial for enhancing the theoretical understanding of PnP/RED methods.
> - **Experimental Expansion:** In response to your suggestion, we have expanded our experimental validation to include CT reconstruction and super-resolution tasks in **Table 2** (see rebuttal to Reviewer 1SLa) and **Table 3**, respectively. These experiments demonstrate the broader applicability and robustness of our method.
> - **Proposition 4.3:** This Proposition provides a theoretical framework for constructing pseudo-contractive denoisers with $d=1$. To practically implement this, it may be necessary to combine the method from reference [20]. We acknowledge that Proposition 4.3 may not seamlessly fit within the paper's focus. We will revise the manuscript to better integrate this proposition and clarify its role in the overall contribution.
>
> ## Reply to Question
> - **A1:** In our experiments, we use the GS denoiser obtained from Proposition 4.1, rather than the denoiser constructed using Proposition 4.3.
> - **A2:** Proposition 4.3 provides a theoretical construction for pseudo-contractive denoisers with $d=1$, potentially providing greater expressive capacity. However, it requires computing the network's Lipschitz constant, which can be challenging. In contrast, the gradient-step denoiser obtained from Proposition 4.1 is a $\frac{L_\theta-2}{L_\theta}$-SPC denoiser. It is practical and easy to implement using existing ICNN architectures, though it has limited expressive capacity due to architectural constraints.
> In summary, denoisers from Proposition 4.3 may be more expressive but less practical, while gradient-step denoisers from Proposition 4.1 are more straightforward to implement but have limited expressive capacity.
> - **A3:** We have included detailed experiments on other inverse problems, such as super-resolution (see **Table 3**) and CT reconstruction (see **Table 2** in rebuttal to Reviewer 1SLa), to demonstrate the versatility and effectiveness of our method.
>
> **Table 3**. Numerical results of different methods on the $\times 2$ super-resolution task with noise level $\sigma=2.55$.  The best two results are highlighted using bold and underline, respectively.
>
> | Methods             | House  | Boats  | Couple | Peppers | Man   |
> |--------------------|--------|--------|--------|---------|-------|
> | PnP-FBS [40]        | 27.76  | 25.35  | 25.08  | 24.13   | 26.17 |
> | RED [2]            | **31.58**  | $\underline{28.35}$ | 27.91  | 26.45   | 29.10 |
> | CRED [12]           | 31.41  | 28.33  | $\underline{27.94}$  | **26.66**   | 29.08 |
> | RED-PRO               | $\underline{31.53}$ | **28.43** | **28.00** | $\underline{26.49}$ | **29.31** |
>
> ## Reply to Limitations
> Thank you for your suggestion. We have provided comparison experiments with state-of-the-art methods in the following **Tables 4** and **Table 5**. These results illustrate the respective advantages and disadvantages, as well as performance gaps, between our approach and other methods. Our findings demonstrate that we can effectively achieve the trade-off between interpretability and performance.
>
> **Table 4**. Deblurring results of different methods on CelebA over 20 samples.
>
> | Method  | PSNR ($\sigma_{blur}=1$, $\sigma_{noise}=.02$) | SSIM ($\sigma_{blur}=1$, $\sigma_{noise}=.02$) | PSNR ($\sigma_{blur}=1$, $\sigma_{noise}=.04$) | SSIM ($\sigma_{blur}=1$, $\sigma_{noise}=.04$) | PSNR ($\sigma_{blur}=2$, $\sigma_{noise}=.02$) | SSIM ($\sigma_{blur}=2$, $\sigma_{noise}=.02$) | PSNR ($\sigma_{blur}=2$, $\sigma_{noise}=.04$) | SSIM ($\sigma_{blur}=2$, $\sigma_{noise}=.04$) |
> |--------------------|-----------------------------------------------|-----------------------------------------------|-----------------------------------------------|-----------------------------------------------|-----------------------------------------------|-----------------------------------------------|-----------------------------------------------|-----------------------------------------------|
> | Blurred and Noisy  | 27.0 ± 1.6  | .80 ± .03 | 24.9 ± 1.0 | .63 ± .05 | 24.0 ± 1.7   | .69 ± .04  | 22.8 ± 1.3  | .54 ± .04 |
> | PnP-BM3D [1] | 31.0 ± 2.7   | .88 ± .04  | 29.5 ± 2.2  | .84 ± .05  | 28.5 ± 2.2  | .82 ± .05 | 27.6 ± 2.0 | .79 ± .05 |
> | PnP-DnCNN [2] | 32.3 ± 2.6 | .90 ± .03  | $\underline{30.9 ± 2.1}$   | .87 ± .04  | 29.5 ± 2.0| .84 ± .04 | 28.3 ± 1.8 | .79 ± .05|
> | DiffPIR [60] | 30.81 ± 2.0 | .86 ± .03  | 29.5 ± 1.8   | .82 ± .03  | 28.6 ± 2.0 | .80 ± .05 | 27.6 ± 1.8 | .77 ± .05|
> | PnP-PGD [28] | 31.4 ± 1.9 | .87 ± .02  | 27.6 ± 0.9   | .71 ± .05  | $\underline{29.9 ± 2.3}$ | .85 ± .05 |$\underline{28.8 ± 2.0}$ | .81 ± .05|
> | DPIR [57] | **33.2± 3.0** | **.92 ± .03**  | **31.75 ± 2.6**| **.89 ± .04**  | **30.1 ± 2.5** | **.86 ± .05** | **29.1 ± 2.2** |**.83 ± .05**|
> | RED-PRO | $\underline{32.4 ± 2.8}$| $\underline{.92 ± .03}$| 30.8 ± 2.3 |$\underline{.88 ± .03}$| 29.3 ± 2.3  |$\underline{.86 ± .04}$|28.4 ± 2.0|$\underline{.83 ± .04}$|
>
> **Table 5**. Comparison with state-of-the-art methods. Test grayscale images are degraded by the Gaussian kernel with the noise level $\sigma = 2.55$.
>
> | Method         | Pepper | Craft | C.man | Couple | Man   | House  | Starfish | Butterfly | Boat   |
> |----------------|--------|-------|-------|--------|-------|--------|----------|-----------|--------|
> | PnP-DRS [29]    | **29.37**| $\underline{26.91}$ | $\underline{27.46}$ | $\underline{29.70}$  | 29.59 | *33.25* | **29.94** | **30.66** | 29.46  |
> | PnP-PGD [28]    | 27.29  | 26.52 | 27.01 | 29.45  | 29.38 | 33.07   | 29.54    | 29.99     | 29.01  |
> | DPIR [57]       | $\underline{28.47}$| **27.02** | **27.63** | **30.24** | **30.89** | **33.56** | $\underline{29.62}$   | $\underline{30.43}$ | **30.54** |
> | RED-PRO| 27.17  | 26.02 | 26.34 | 29.17  | $\underline{30.25}$ | 32.40   | 28.14    | 28.75 | $\underline{29.54}$ |

---

### Official Review · Reviewer_1SLa · 2025-07-01

**Clarity:** 3
**Significance:** 3
**Originality:** 3
**Rating:** 5
**Confidence:** 3

**Summary:**

This method combines gradient step denoisers with input convex neural networks (ICNNs) to structurally ensure the pseudo-contractive property. It avoids additional spectral norm penalties while guaranteeing the theoretical convergence of the RED-PRO model. Experiments demonstrate stable performance in image deblurring tasks, balancing interpretability and effectiveness.

**Questions:**

1. The experiments are primarily conducted on relatively simple datasets such as MNIST and CelebA. Would the method still perform effectively on more complex or diverse datasets, such as ImageNet or real-world natural scenes?

2. Could the authors consider including experiments in the domain of medical image reconstruction, such as MRI or CT, to further demonstrate the generalizability of their method to high-impact inverse problems?

**Ethical Concerns:**

["NO or VERY MINOR ethics concerns only"]

**Final Justification:**

The authors provided additional experiments on CT. The scale of the experiments is relatively small, and as other reviewers have pointed out, the compared algorithms are also relatively simple. I would like to keep the current score.

**Limitations:**

yes

**Quality:**

4

**Strengths And Weaknesses:**

Strengths:
1. The paper is the first to prove that gradient step denoisers parameterized by ICNNs are inherently d-strictly pseudo-contractive.
2. The learned SPC denoiser is integrated into the RED-PRO framework, and both sequence and objective convergence are established.
3. Experiments demonstrate that the learned denoiser strictly satisfies the SPC condition across noise levels and achieves competitive or superior performance in image deblurring compared to PnP-BM3D and PnP-DnCNN.

Weaknesses:
1. The denoiser is constructed using ICNNs) to ensure convexity and, consequently, the pseudo-contractive property. However, due to the architectural constraints of ICNNs—specifically the requirement for non-negative weights and monotonic activation functions—the expressive capacity of the model is inherently limited. This may hinder its ability to capture complex image structures or non-convex priors, especially in high-complexity tasks such as natural image denoising or super-resolution.
2. Although the proposed method avoids frequent power iteration steps used in spectral normalization, it still requires estimating the global Lipschitz constant of the denoising network.
3. The experiments are conducted on relatively simple datasets such as MNIST and CelebA. While CelebA contains real facial images, the overall image diversity and structural complexity are limited.

---

> ### Author Rebuttal · Authors · 2025-07-30
>
> # To Reviewer 1SLa
> We sincerely appreciate your insightful feedback and positive evaluation of our work. We have addressed your concerns regarding theoretical guarantees, experimental validation, and practical applicability as follows:
> ## Reply to Weaknesses
> - **Expressive Capacity of ICNNs:** We acknowledge that the non-negative weight and monotonic activation constraints may limit the model’s ability to capture highly complex image structures or non-convex priors. Our primary goal is to ensure the truly pseudo-contractive property for theoretical guarantees, which we believe is a significant contribution. And we believe the trade-off between interpretability and expressiveness is a promising direction for future research.
> - **Lipschitz Constant Estimation:** While our method circumvents frequent power iterations, the estimation of the global Lipschitz constant remains challenging. We are actively exploring efficient estimation techniques and plan to incorporate these in our future work.
> - **Experiments Complexity:** We selected MNIST and CelebA to demonstrate the theoretical and practical feasibility of our approach. We agree that evaluating on more complex and diverse datasets would strengthen our findings. We have already included comparisons with various methods on natural images in **Table 1**.
>
> ## Reply to Question
> **A1.** We agree that evaluating our method on more complex and diverse datasets, such as ImageNet or real-world natural scenes, would provide a more compelling demonstration of its effectiveness.
>
> We train an FNE DnCNN using the spectral method [40], which is employed in PnP-FBS and RED, as detailed in the following **Table 1**. In contrast, RED-PRO uses the GS denoiser described in Proposition 4.1. The quantitative results in **Table 1** show that RED-PRO outperforms comparable methods on the House and Butterfly images.
>
> **Table 1**. Numerical results of different algorithms on the deblurring task with a Gaussian blur kernel ($\sigma_{\mathbf{A}} = 1.6$) and noise level $\sigma =2.55$. The best two results are highlighted using bold and underline, respectively.
>
> | Methods               | House  | Boats  | Butterfly | Couple | Peppers |
> |---------------------|--------|--------|-----------|--------|---------|
> | PnP-FBS [40]         | 30.49  | 27.36  | 25.51    | 26.90  | 26.00   |
> | RED [2]             | $\underline{32.36}$  | **29.56**  | $\underline{28.35}$ | **29.19**  | $\underline{27.18}$ |
> | CRED [12]            | 32.28  | 29.44  | 28.70     | 29.11  | **27.46**   |
> | RED-PRO                | **32.40**  | $\underline{29.54}$  | **28.75** | $\underline{29.17}$  | 27.17 |
>
> **A2.** We also appreciate your suggestion to explore medical image reconstruction tasks, such as MRI or CT. We consider a sparse-view computed tomography (CT) measurement model defined as follows:
> $$ \min_{\mathbf{x}\in \mathrm{Fix}(T_\theta)} \frac{1}{N}\sum_{i=1}^N || \mathbf{A}_i \mathbf{x}-\mathbf{b}_i ||^2,
> $$
> where $\mathbf{b}_i \in \mathbb{R}^m$ is the measured sinogram for the $i$-th projection, and $\mathbf{A}_i$ is an $m \times n$ discretized Radon transform matrix. For RED-PRO, we set the paremeter $\mu_k =\frac{2}{|| \mathbf{A} ||^2(1+k)^{0.01}}$, where $\mathbf{A} =[\mathbf{A}_1,\mathbf{A}_2,\ldots, \mathbf{A}_N]^\mathrm{T}$, and $w=0.1$. The GS denoiser is trained on the publicly available Mayo-CT dataset. We simulate CT sinograms using a parallel-beam geometry with 200 angles and 400 detectors. The results for CT reconstruction are presented in **Table 2**.
>
> **Table 2**. Numerical results for CT reconstruction
> on the Mayo-CT dataset, computed over 128 test images.
>
> | Method | PSNR                | SSIM                |
> |--------|---------------------|---------------------|
> | FBP    | 20.233 ± 0.034      | 0.1763 ± 0.0138     |
> | RED-PRO   | 30.057 ± 0.488      | 0.8190 ± 0.0075     |

---

> > ### Comment · Reviewer_1SLa · 2025-08-06
> >
> > Thank you for your rebuttal. After considering the disscussion, I have decided to keep the current score.

---

### Decision · Program_Chairs · 2025-09-17

**Decision:**

Accept (poster)

**Comment:**

This paper studies denoising algorithms for general inverse problems, focusing on the RED (and RED-PRO) approach. In this setting, it is important for the employed denoisers to be pseudo-contractive. While other partially successful methods exist, the authors show here that gradient-step (GS) denoisers (introduced not long ago and parametrized by ICNNs) are indeed pseudo-contractive. With this results, the authors then integrate (GS) within RED-PRO, with the respective convergence guarantees. Experimental results are provided on image deblurring.

**Strengths**
- The result presented in the paper (linking GS denoisers with pseudo-contractivity) is new and interesting.
- Nice and elegant integration with RED-PRO with provable guarantees.
- Experiments validate their SPC property, and show compelling results.

**Weaknesses**
- By constraining the denser to be a GS (parametrized by an ICCN), the performance of the model is limited (unsurprisingly).
- While the result is novel, it follows rather straightforwardly from known results.
- There were some inconsistencies on the technical details, raised by Reviewer Ygzw’s.
- The experimental study was limited to a single inverse problem (deblurring) and on a simple datasets.

**Discussion and conclusion**

The discussions between reviewers and authors was productive. The authors have included further experimental results on two other tasks (CT reconstruction and super-resolution), including other (newer and stronger) methods for comparison, significantly expanding on their numerical component. The new results indicate that this method trades performance for theoretical guarantees, as expected. The technical details and innovations that were not completely clear were clarified by the authors, and references and clarification comments were (promised to be) added.

Three reviewers recommend acceptance (two strong, one moderate), while one reviewer maintains a weak reject on concerns over technical rigor and experimental sufficiency. If the reviewers implement the changes that were raised through the rebuttal process, I am confident that the technical rigor and the experimental section will be much improved, and that this paper will be an interesting addition to the community.